# Global Convergence of Federated Learning for Mixed Regression

**Lili Su**
Electrical and Computer Engineering
Northeastern University
l.su@northeastern.edu

**Jiaming Xu**
The Fuqua School of Business
Duke University
jiaming.xu868@duke.edu

**Pengkun Yang**
Center for Statistical Science
Tsinghua University
yangpengkun@tsinghua.edu.cn

## Abstract

This paper studies the problem of model training under Federated Learning when clients exhibit cluster structure. We contextualize this problem in mixed regression, where each client has limited local data generated from one of $k$ unknown regression models. We design an algorithm that achieves global convergence from any initialization, and works even when local data volume is highly unbalanced – there could exist clients that contain $O(1)$ data points only. Our algorithm first runs moment descent on a few anchor clients (each with $\tilde{\Omega}(k)$ data points) to obtain coarse model estimates. Then each client alternately estimates its cluster labels and refines the model estimates based on FedAvg or FedProx. A key innovation in our analysis is a uniform estimate on the clustering errors, which we prove by bounding the VC dimension of general polynomial concept classes based on the theory of algebraic geometry.

## 1 Introduction

Federated learning (FL) [MMR+17] enables a massive number of clients to collaboratively train models without disclosing raw data. Heterogeneity in clients renders local data non-IID and highly unbalanced. For example, smartphone users have different preferences in article categories (e.g. politics, sports or entertainment) and have a wide range of reading frequencies. In fact, distribution of the local dataset sizes is often heavy-tailed [DDPM13, FAC, FHK12].

Existing methods to deal with data heterogeneity can be roughly classified into three categories: a common model, fully personalized models, and clustered personalized models; see Section 2 for detailed discussion. Using a common model to serve highly heterogeneous clients has fundamental drawbacks; recent work [SXY21] rigorously quantified the heterogeneity level that the common model can tolerate with. Fully personalized models [SCST17, MNB+21] often do not come with performance guarantees as the underlying optimization problem is generally hard to solve. In this work, we focus on clustered personalized models [SMS20], i.e., clients within the same cluster share the same underlying model and clients across clusters have relatively different underlying models. The main challenge is that the cluster identities of the clients are unknown. We target at designing algorithms that simultaneously learn the clusters and train models for each cluster.

A handful of simultaneous clustering and training algorithms have been proposed [MMRS20, XLS+21, LLV21, GCYR20], mostly of which are heuristic and lack of convergence guarantees

[MMRS20, XLS⁺21, LLV21]. Towards formal assurance, [GCYR20] studied this problem through the lens of statistical learning yet postulated a number of strong assumptions such as the initial model estimates are very close to the true ones, linear models, strong convexity, balanced and high-volume of local data. Their numerical results [GCYR20] suggested that sufficiently many random initializations would lead to at least one good realization satisfying the required closeness assumption. However, the necessary number of random initializations scales exponentially in both the input dimension and the number of clusters. Besides, in practice, it is hard to recognize and winnow out good initialization. In this work, following [GCYR20], we adopt a statistical learning setup. In particular, we contextualize our problem as the canonical mixed regression, where each client has a set of local data generated from one of $k$ unknown regression models. Departing from standard mixed regression, in which each client keeps one data point only [MN19, LL18], in our problem the sizes of local datasets can vary significantly across clients. We only make a mild assumption that there exist a few anchor clients (each with $\tilde{\Omega}(k)$ data points) and sufficiently many clients with at least two data points. Similar mixed regression setup with data heterogeneity has been considered in [KSS⁺20] in a different context of meta-learning; the focus there is on exploring structural similarities among a large number of tasks in centralized learning. On the technical side, their analysis only works when the covariance matrices of all the clusters are identical and each client has $\Omega(\log k)$ data points. Please refer to Remark 1 for more detailed technical comparisons.

**Contributions**  The main contributions of this work are summarized as follows:

- We design a two-phase federated learning algorithm to learn clustered personalized models in the context of mixed regression problems. In Phase 1, the parameter server runs a federated moment descent on a few anchor clients to obtain coarse model estimates based on subspace estimation. In each global iteration of Phase 2, each client alternately estimates its cluster label and refines the model estimates based on FedAvg or FedProx. The algorithm works even when local data volume is highly unbalanced – there could exist clients that contain $O(1)$ data points only.

- We prove the global convergence of our algorithm from any initialization. The proof is built upon two key ingredients: 1) We develop a novel eigengap-free bound to control the projection errors in subspace estimation; 2) To deal with the sophisticated interdependence between the two phases and across iterations, we develop a novel uniform estimate on the clustering errors, which we derive by bounding the VC dimension of general polynomial concept classes based on the theory of algebraic geometry. Our analysis reveals that the final estimation error is dominated by the uniform deviation of the clustering errors, which is largely overlooked by the previous work.

## 2 Related Work

FedAvg [MMR⁺17] is a widely adopted FL algorithm due to its simplicity and low communication cost. However, severe data heterogeneity could lead to unstable training trajectories and land in suboptimal models [LSZ⁺20, ZLL⁺18, KKM⁺20]. Based on the number of models trained, existing methods to deal with data heterogeneity can be roughly classified into three categories.

**A Common Model:** To limit the negative impacts of data heterogeneity on the obtained common model, a variety of techniques based on variance reduction [LSZ⁺20, LXC⁺19, KKM⁺20] and normalization [WLL⁺20] have been introduced. Their convergence results mostly are derived under strong technical assumptions such as bounded gradient and/or bounded Hessian dissimilarity which do not hold when the underlying truth in the data generation is taken into account [LSZ⁺20, LXC⁺19, KKM⁺20]. In fact, none of them strictly outperform others in different instances of data heterogeneity [LDCH21]. Besides, the generalization errors of the common model with respect to local data are mostly overlooked except for a recent work [SXY21], which shows that the common model can tolerate only a moderate level of model heterogeneity.

**Fully Personalized Models:** [SCST17] proposed Federated Multi-Task Learning (MTL) wherein different models are learned for each of the massive population of clients [SCST17, MNB⁺21]. Though conceptually attractive, the convergence behaviors of Federated MTL is far from well-understood because the objective is not jointly convex in the model parameters and the model relationships [SCST17, AEP08, AZB05]. Specifically, [SCST17] focused on solving the subproblem

of updating the model parameters only. Even in the centralized setting, convergence is only shown under rather restricted assumptions such as equal dataset sizes for different tasks (i.e. balanced local data) [AZB05] and small number of common features [AEP08]. Moreover, the average excess error rather than the error of individual tasks is shown to decay with the dominating term $O(1/\sqrt{n})$, where $n$ is the homogeneous local dataset size [AZB05]. Despite recent progress [TJJ21, DHK$^+$20], their results are mainly for linear representation learning and for balanced local data. Parallel to Federated MTL, model personalization is also studied under the Model-Agnostic Meta-Learning (MAML) framework [FMO20, JKRK19] where the global objective is modified to account for the cost of fine-tuning a global model at individual clients. However, they focused on studying the convergence in training errors only– no characterization of the generalization error is given.

**Clustered Personalized Models:** Clustered Federated Learning (CFL) [SMS20, GCYR20, GHYR19, XLS$^+$21, MMRS20, LLV21] can be viewed as a special case of Federated MTL where tasks across clients form cluster structures. In addition to the algorithms mentioned in Section 1, i.e., which simultaneously learn clusters and models for each cluster, other attempts have been made to integrate clustering with model training. [SMS20] hierarchically clustered the clients in a post-processing fashion. To recover the $k$ clusters, $\Omega(k)$ empirical risk minimization problems need to be solved sequentially – which is time-consuming. [GHYR19] proposed a modular algorithm that contains one-shot clustering stage, followed by $k$ individual adversary-resilient model training. Their algorithm scales poorly in the input dimension, and requires local datasets to be balanced and sufficiently large (i.e., $n \geq d^2$). Moreover, each client sequentially solves two empirical risk minimization problems. To utilize information across different clusters, [LLV21] proposed soft clustering. Unfortunately, the proposed algorithm is only tested on the simple MNIST and Fashion-MNIST datasets, and no theoretical justifications are given.

## 3 Problem Formulation

A FL system consists of a parameter server (PS) and $M$ clients. Each client $i \in [M]$ keeps a dataset $\mathcal{D}_i = \{(x_{ij}, y_{ij})\}_{j=1}^{n_i}$ that are generated from one of $k$ unknown regression models. Let $N = \sum_{i=1}^M n_i$. The local datasets are highly unbalanced with varying $n_i$ across clients. If $n_i = \tilde{\Omega}(k)$, we refer to client $i$ as *anchor* client, which corresponds to active user in practice. Anchor clients play a crucial rule in our algorithm design. We consider the challenging yet practical scenario wherein a non-anchor client may have $O(1)$ data points only.

We adopt a canonical mixture model setup: For each client $i \in [M]$,

$$y_i = \phi(\boldsymbol{x}_i)\theta_{z_i}^* + \zeta_i, \tag{1}$$

where $z_i \in [k]$ is the *hidden* local cluster label, $\theta_1^*, \cdots, \theta_k^*$ are the true models of the clusters, $\phi(\boldsymbol{x}_i) \in \mathbb{R}^{n_i \times d}$ is the feature matrix with rows given by $\phi(x_{ij})$, $y_i = (y_{ij}) \in \mathbb{R}^{n_i}$ is the response vector, and $\zeta_i = (\zeta_{ij}) \in \mathbb{R}^{n_i}$ is the noise vector. Note that the feature map $\phi$ can be non-linear (e.g. polynomials). The cluster label of client $i$ is randomly generated from one of the $k$ components from some unknown $p = (p_1, \ldots, p_k)$ in probability simplex $\boldsymbol{\Delta}^{k-1}$. That is, $\mathbb{P}\{z_i = j\} = p_j$ for $j \in [k]$. In addition, $\|\theta_j^*\|_2 \leq R$ for each component. The feature covariate $\phi(x_{ij})$ is independent and sub-Gaussian $\alpha I_d \preceq \mathbb{E}[\phi(x_{ij})\phi(x_{ij})^\top] \preceq \beta I_d$. We assume that the covariance matrix is identical within the same cluster but may vary across different clusters, i.e., $\mathbb{E}[\phi(x_{ij})\phi(x_{ij})^\top] = \Sigma_j$ if $z_i = j$. The noise $\zeta_{ij}$ is independent and sub-Gaussian with $\mathbb{E}[\zeta_{ij}] = 0$ and $\mathbb{E}[\zeta_{ij}^2] \leq \sigma^2$.

Our formulation accommodates statistical heterogeneity in feature covariates, local models, and observation noises [KMA$^+$21]. For the identifiability of the true cluster models $\theta_j^*$'s, we assume a minimum proportion and a pairwise separation of clusters. Formally, let $\Delta = \min_{j \neq j'} \|\theta_j^* - \theta_{j'}^*\|_2$ and $p_{\min} = \min_{j \in [k]} p_j$. For ease of presentation, we assume the parameters $\alpha, \beta = \Theta(1), \sigma/\Delta = O(1)$, and $R/\Delta = O(1)$, while our main results can be extended to show more explicit dependencies on these parameters with careful bookkeeping calculations. Note that even under these assumptions, we still allow $R, \Delta, \sigma$ to scale with model dimension $d$. Also, the assumption $R/\Delta = O(1)$ basically requires the radius of $\theta_j^*$'s is on the same scale as their pairwise separation. It rules out the extreme setting where $\theta_j^*$'s themselves are extremely large while their pairwise separations are tiny.

**Notations:** Let $[n] \triangleq \{1, \ldots, n\}$. For two sets $A$ and $B$, let $A \ominus B$ denote the symmetric difference $(A - B) \cup (B - A)$. We use standard asymptotic notation: for two positive sequences $\{a_n\}$ and

$\{b_n\}$, we write $a_n = O(b_n)$ (or $a_n \lesssim b_n$) if $a_n \leq Cb_n$ for some constant $C$ and sufficiently large $n$; $a_n = \Omega(b_n)$ (or $a_n \gtrsim b_n$) if $b_n = O(a_n)$; $a_n = \Theta(b_n)$ (or $a_n \asymp b_n$) if $a_n = O(b_n)$ and $a_n = \Omega(b_n)$; Poly-logarithmic factors are hidden in $\tilde{\Omega}$. Given a matrix $A \in \mathbb{R}^{n \times d}$, let $A = \sum_{i=1}^r \sigma_i u_i v_i^\top$ denote its singular value decomposition, where $r = \min\{n, d\}$, $\sigma_1 \geq \cdots \geq \sigma_r \geq 0$ are the singular values, and $u_i$ ($v_i$) are the corresponding left (right) singular vectors. We call $U = [u_1, u_2, \ldots, u_k]$ as the top-$k$ left singular matrix of $A$. Let $\text{span}(U) = \text{span}\{u_1, \ldots, u_k\}$ denote the $k$-dimensional subspace spanned by $\{u_1, \ldots, u_k\}$.

# 4 Main Results

We propose a two-phase FL algorithm that enables clients to learn the model parameters $\theta_1^*, \ldots, \theta_k^*$ and their clusters simultaneously:

*(i) Coarse estimation via FedMD.* Run the federated moment descent algorithm to obtain coarse estimates of model parameters $\theta_i^*$'s.

*(ii) Fine-tuning via iterative FedX+clustering.* In each iteration, each client first estimates its cluster label and then refines its local model estimate via either FedAvg or FedProx (which we refer to as FedX) [MMR+17, LSZ+20].

FedMD and FedX+clustering are detailed in the pseudocode in Phase 1 and Phase 2, respectively.

## 4.1 Federated moment descent

With multiple clusters and sub-Gaussian features, simple procedures such as power method will no longer provide a reasonably good coarse estimation. The reasons are two-fold: 1) With sub-Gaussian features, it is difficult to construct a matrix whose leading eigenspace approximately aligns with the space spanned by the true model parameters $(\theta_1^*, \ldots, \theta_k^*)$; 2) Even this is achievable, there still remains significant ambiguity in determining the model parameters from their spanned subspace.

The key idea of the first phase of our algorithm is to leverage the existence of anchor clients. Specifically, the PS chooses a set $H$ of $n_H$ anchor clients uniformly at random. Each selected anchor client $i \in H$ maintains a sequence of estimators $\{\theta_{i,t}\}$ that approaches $\theta_{z_i}^*$, achieving $\|\theta_{i,t} - \theta_{z_i}^*\|_2 \leq \epsilon \Delta$ for some small constant $\epsilon > 0$ when $t$ is sufficiently large.

At high-level, we hope to have $\theta_{i,t}$ move along a well calibrated direction $r_{i,t}$ that decreases the residual estimation error $\|\Sigma_{z_i}^{1/2}(\theta_{z_i}^* - \theta_{i,t})\|_2^2$, i.e., the variance of the residual $\langle \phi(x_{ij}), \theta^* - \theta_{i,t}\rangle$. As such, we like to choose $r_{i,t}$ to be positively correlated with $\Sigma_{z_i}(\theta_{z_i}^* - \theta_{i,t})$. However, to estimate $\Sigma_{z_i}(\theta_{z_i}^* - \theta_{i,t})$ solely based on the local data of anchor client $i$, it requires $n_i = \tilde{\Omega}(d)$, which is unaffordable in typical FL systems with high model dimension $d$ and limited local data. To resolve the curse of dimensionality, we decompose the estimation task at each chosen anchor client into two subtasks: we first estimate a $k$-dimensional subspace that $\Sigma_{z_i}(\theta_{z_i}^* - \theta_{i,t})$ lies in by pooling local datasets across sufficiently many non-anchor clients; then we project the local data of anchor client $i$ onto the estimated subspace and reduce the estimation problem from $d$-dimension to $k$-dimension.

The precise description of our Phase 1 procedure is given as below. For ease of notation, let $\varepsilon(x, y, \theta) \triangleq (y - \langle \phi(x), \theta\rangle)\phi(x)$.

In Step 9, PS estimates the subspace that the residual estimation errors $\{\Sigma_j(\theta_j^* - \theta_{i,t})\}_{j=1}^k$ lie in, in collaboration with clients in $\mathcal{S}_t$. In particular, for each anchor client $i \in H$, define

$$Y_{i,t} = \frac{1}{m} \sum_{i' \in \mathcal{S}_t} \varepsilon(x_{i'1}, y_{i'1}, \theta_{i,t}) \varepsilon(x_{i'2}, y_{i'2}, \theta_{i,t})^\top$$

We approximate the subspace spanned by $\{\Sigma_j(\theta_j^* - \theta_{i,t})\}_{j=1}^k$ via that spanned by the top-$k$ left singular vectors of $Y_{i,t}$. To compute the latter, we adopt the following multi-dimensional generalization of the power method, known as orthogonal-iteration [GVL13, Section 8.2.4]. In general, given a symmetric matrix $Y \in \mathbb{R}^{d \times d}$, the orthogonal iteration generates a sequence of matrices $Q_t \in \mathbb{R}^{d \times k}$ as follows: $Q_0 \in \mathbb{R}^{d \times k}$ is initialized as a random orthogonal matrix $Q_0^\top Q_0 = I$ and $Y Q_t = Q_{t+1} R_{t+1}$ with QR factorization. When $t$ is large, $Q_t$ approximates the top-$k$ left singualr matrix of $Y$, provided the existence of an eigen-gap $\lambda_k > \lambda_{k+1}$. When $k = 1$, this is just the the

---

**Phase 1:** Federated Moment Descent (FedMD)

1 **Input:** $n_H, k, m, \ell, T, T_1, T_2 \in \mathbb{N}, \alpha, \beta, \epsilon, \Delta \in \mathbb{R}, \theta_0 \in \mathbb{R}^d$ with $\|\theta_0\|_2 \le R$

2 **Output:** $\hat{\theta}_1, \ldots, \hat{\theta}_k$

3 PS chooses a set $H$ of $n_H$ anchor clients uniformly at random;

4 **for** *each anchor client $i \in H$* **do**

5 $\quad \lfloor \quad \theta_{i,0} \leftarrow \theta_0$;

6 **for** $t = 0, 1, \ldots, T-1$ **do**

7 $\quad$ PS selects a set $\mathcal{S}_t$ of $m$ clients from $[M] \setminus \left( H \cup \left( \cup_{\tau=0}^{t-1} \mathcal{S}_\tau \right) \right)$;

8 $\quad$ PS broadcasts $\{\theta_{i,t}, i \in H\}$ to all clients $i'$ in $\mathcal{S}_t$; /* where $\cup_{\tau=0}^{-1} \mathcal{S}_\tau = \emptyset$ */;

9 $\quad$ PS calls federated-orthogonal-iteration $(\mathcal{S}_t, \{\varepsilon(x_{i'1}, y_{i'1}, \theta_{i,t}), \varepsilon(x_{i'2}, y_{i'2}, \theta_{i,t})\}_{i' \in \mathcal{S}_t},$
$\quad\quad k, T_1)$ to output $\hat{U}_{i,t}$ for each anchor client $i \in H$; /* described
$\quad\quad$ in Algorithm 3 */

10 $\quad$ PS sends $\hat{U}_{i,t}$ to each anchor client $i \in H$;

11 $\quad$ Each anchor client $i$ calls power-iteration$(A_{i,t} A_{i,t}^\top, T_2)$ to output $(\hat{\beta}_{i,t}, \hat{\sigma}_{i,t}^2)$ with $A_{i,t}$
$\quad\quad$ defined in (2);

12 $\quad$ **if** $\hat{\sigma}_{i,t} > \epsilon \Delta$ **then**

13 $\quad\quad\quad \theta_{i,t+1} \leftarrow \theta_{i,t} + r_{i,t} \eta_{i,t}$ and reports $\theta_{i,t+1}$, where $r_{i,t} = \hat{U}_{i,t} \hat{\beta}_{i,t}$ and
$\quad\quad\quad \eta_{i,t} = \alpha \hat{\sigma}_{i,t} / (2\beta^2)$;

14 $\quad$ **else**

15 $\quad\quad\quad \lfloor \quad \theta_{i,t+1} \leftarrow \theta_{i,t}$ and reports $\theta_{i,t+1}$;

16 PS computes the pairwise distance $\|\theta_{i,T} - \theta_{i',T}\|_2$ for every pair of anchor clients $i, i' \in H$,
assigns them in the same cluster when the pairwise distance is smaller than $\Delta/2$, and
outputs $\hat{\theta}_j$ to be the center of the estimated $j$-th cluster for $j \in [k]$.

---

power-iteration and we can further approximate the leading eigenvalue of $Y$ by the Raleigh quotient $Q_t^\top Y Q_t$. When $Y$ is asymmetric, by running the orthogonal iteration on $YY^\top$, we can compute the top-$k$ left singular matrix of $Y$. In our setting, the orthogonal iteration can be implemented in a distributed manner in FL systems as shown in Algorithm 3 in the Appendix D.1.

In Step 11, each anchor client $i$ estimates the residual error $\Sigma_{z_i}(\theta_{z_i}^* - \theta_{i,t})$ by projecting $\varepsilon(x_{ij}, y_{ij}, \theta_{i,t})$ onto the previously estimated subspace, that is, $\hat{U}_{i,t}^\top \varepsilon(x_{ij}, y_{ij}, \theta_{i,t})$. This reduces the estimation from $d$-dimension to $k$-dimension and hence $\tilde{\Omega}(k)$ local data points suffice. Specifically, define

$$A_{i,t} = \frac{1}{\ell} \sum_{j \in \mathcal{D}_{i,t}} \left( \hat{U}_{i,t}^\top \varepsilon(x_{ij}, y_{ij}, \theta_{i,t}) \right) \left( \hat{U}_{i,t}^\top \varepsilon(\tilde{x}_{ij}, \tilde{y}_{ij}, \theta_{i,t}) \right)^\top, \tag{2}$$

where $\mathcal{D}_{i,t}$ consists of $2\ell$ local data points $(x_{ij}, y_{ij})$ and $(\tilde{x}_{ij}, \tilde{y}_{ij})$ freshly drawn from $\mathcal{D}_i$ at iteration $t$. Client $i$ runs the power-iteration to output $\hat{\beta}_{i,t}$ and $\hat{\sigma}_{i,t}^2$ as approximations of the leading left singular vector and singular value of $A_{i,t}$. Then anchor client $i$ updates $\theta_{i,t+1}$ by moving along the direction of the estimated residual error $r_{i,t}$ with an appropriately chosen step size $\eta_{i,t}$.

We show that $\theta_{i,T}$ is close to $\theta_{z_i}^*$ for every anchor client $i \in H$ and the outputs $\hat{\theta}_j$ are close to $\theta_j^*$ up to a permutation of cluster indices.

**Theorem 1.** *Let $\epsilon \in (0, 1/4)$ be a small but fixed constant. Suppose that*

$$m \ge p_{\min}^{-2} \tilde{\Omega}(d), \ \ell = \tilde{\Omega}(k), \ T = \Omega(1), \ T_1 = \Omega(k \log(Nd)), \ T_2 = \Omega(\log(Nd)). \tag{3}$$

*With probability at least $1 - O(n_H T / N^{10})$, for all initialization $\theta_0$ with $\|\theta_0\|_2 \le R$,*

$$\sup_{i \in H} \|\theta_{i,T} - \theta_{z_i}^*\|_2 \le \epsilon \Delta. \tag{4}$$

*Furthermore, when $n_H \ge \log N / p_{\min}$, with probability at least $1 - O(n_H T / N^{10})$,*

$$d(\hat{\theta}, \theta^*) \le \epsilon \Delta, \tag{5}$$

*where*

$$d(\hat{\theta}, \theta^*) \triangleq \min_{\pi} \max_{j \in [k]} \left\| \hat{\theta}_{\pi(j)} - \theta_j^* \right\|_2, \quad \text{where } \pi \text{ is permutation over } [k]. \tag{6}$$

Note that in (6) we take a minimization over permutation, as the cluster indices are unidentifiable.

Phase 1 uses fresh data at every iteration. In total we need $p_{\min}^{-2} \tilde{\Omega}(d)$ clients with at least two data points and $\tilde{\Omega}(1/p_{\min})$ anchor clients. This requirement is relatively mild, as typical FL systems have a large number of clients with $O(1)$ data points and a few anchor clients with moderate data volume.

We defer the detailed proof of Theorem 1 to Appendix D. A key step in our proof is to show the residual estimation errors $\{\Sigma_j(\theta_j^* - \theta_{i,t})\}_{j=1}^k$ approximately lie in $\text{span}(\hat{U}_{i,t})$. Unfortunately, the eigengap of $Y_{i,t}$ could be small, especially when $\theta_{i,t}$ gets close to $\theta_{z_i}^*$; and hence the standard Davis-Kahan theorem [DK70] cannot be be applied. This issue is further exaggerated by the fact that the convergence rate of the orthogonal iteration also crucially depends on the eigengaps [GVL13]. For these reasons, $\text{span}(\hat{U}_{i,t})$ may not be close to $\text{span}\{\Sigma_j(\theta_j^* - \theta_{i,t})\}_{j=1}^k$ at all. To resolve this issue, we develop a novel gap-free bound to show that projection errors $\hat{U}_{i,t}^\top \Sigma_j(\theta_j^* - \theta_{i,t})$ are small for every $j \in [k]$ (cf. Lemma 5).

**Remark 1** (Comparison to previous work [LL18, KSS+20]). Our algorithm is partly inspired by [LL18] which focuses on the noiseless mixed linear regression, but deviates in a number of crucial aspects. First, our algorithm crucially utilizes the fact that each client chosen in $\mathcal{S}_t$ has at least two data points and hence the space of the singular vectors of $\mathbb{E}[Y_{i,t}]$ is spanned by $\{\Sigma_j(\theta_j^* - \theta_{i,t})\}_{j=1}^k$. In contrast, [LL18] relies on the sophisticated method of moments which only works under the Gaussian features and requires exponential in $k^2$ many data points. Second, our algorithm crucially exploits the existence of anchor clients and greatly simplifies the moment descent algorithm in [LL18].

Our algorithm also bears similarities with the meta-learning algorithm in [KSS+20], which also uses clients collectively for subspace estimation and anchor clients for estimating cluster centers. However, there are several key differences. First, [KSS+20] focuses on the centralized setting and relies on one-shot estimation, under the additional assumption that the covariance matrix of features across all clusters are identical. Instead, our moment descent algorithm is iterative, is amenable to a distributed implementation in FL systems, and allows for covariance matrices varying across clusters. Second, in the fine-tuning phase, [KSS+20] uses the centralized least squares to refine the clusters estimated with anchor clients, under the additional assumption that $\Omega(\log k)$ data points for every client. In contrast, as we will show later, we use the FedX+clustering to iteratively cluster clients and refine cluster center estimation.

**Remark 2.** (Data privacy risk) Compared to the standard FedAvg algorithm wherein only aggregated local updates/gradients are broadcasted by the parameter server, the major step of our two-phase algorithm that may leak additional privacy is Step 8 wherein the local model estimates of the anchor clients are broadcasted to many other non-anchor clients. However, this privacy leakage is minor and can be further mitigated by a simple privacy-preserving mechanism according to the following considerations.

First, in our algorithm, each chosen non-anchor client only receives a collection of local model estimates (without ID for anchor clients) from the parameter server, it does not know which broadcasted model corresponds to which anchor client and hence cannot directly identify each individual anchor client's local true model. Second, we only choose a very few number of anchor clients (roughly on the order of the number of clusters) and in practice these anchor clients are often specially recruited by the PS; hence they can be made less concerned about privacy leakage through some incentivizing schemes. Last but not least, we can better preserve the privacy of anchor clients by broadcasting perturbed versions of their local models to each client. Specifically, fix any anchor client $i$, each non-anchor client $i'$ receives $\theta_{i',i,t}$ and $\tilde{\theta}_{i',i,t}$ that are equal to $\theta_{i,t}$ subject to two independent noise perturbations. Then for the subspace estimation in Step 9, we can replace one $\theta_{i,t}$ by $\theta_{i',i,t}$ and the other by $\tilde{\theta}_{i',i,t}$, in the definition of $Y_{i,t}$. Crucially, $Y_{i,t}$ involves an average over $m$ non-anchor clients; hence these independent noise perturbations for different $i'$ will be averaged out. Since $m$ is large, this implies that the injected random noises can be made large without deteriorating too much the accuracy of the subspace estimation, in a similar spirit as privatizing the model averaging step in FedAvg. This gives a promising pathway to maintain anchor clients' privacy; we leave rigorously analyzing its privacy guarantee as future work.

## 4.2 FedX+clustering

At the end of Phase 1, only the selected anchor clients in $H$ obtained coarse estimates of their local true models (characterized in (4)). In Phase 2, both anchor clients in $H$ and all the other clients (anchor or not) will participate and update their local model estimates.

Phase 2 is stated in a generic form for any loss function $L(\theta, \lambda; \mathcal{D})$, where $\theta = (\theta_1, \ldots, \theta_k) \in \mathbb{R}^{dk}$ is the cluster parameters, $\lambda \in \mathbf{\Delta}^{k-1}$ represents the likelihood of the cluster identity of a client, and $\mathcal{D}$ denotes the client's dataset. This generic structure covers the idea of soft clustering [LLV21]. Note that unlike Phase 1 where each anchor client $i$ only maintains an estimate $\theta_{i,t}$ of its own model, in Phase 2, each client $i$ maintains model estimates $\theta_{i\cdot,t} = (\theta_{i1,t}, \ldots, \theta_{ik,t})$ for all clusters.

---

**Phase 2:** FedX+clustering

---

**1** **Input:** $\theta = (\theta_1, \ldots, \theta_k)$ from the output of Phase 1, $\eta, T'$.

**2** **Output:** $\hat{\theta} = (\hat{\theta}_1, \ldots, \hat{\theta}_k)$

**3** PS sets $\theta_T \leftarrow \theta$.

**4** **for** $t = T+1, \ldots, T+T'$ **do**

**5** $\quad$ PS broadcasts $\theta_{t-1}$ to all clients;

**6** $\quad$ Each client $i$ estimates the likelihood of its local cluster label by

$$\lambda_{i,t} \leftarrow \arg\min_{\lambda \in \mathbf{\Delta}^{k-1}} L(\theta_{t-1}, \lambda; \mathcal{D}_i); \tag{7}$$

**7** $\quad$ Each client $i$ refines its local model based on either FedAvg or FedProx with $L_i(\theta) = L(\theta, \lambda_{i,t}; \mathcal{D}_i)$, and reports the updated local parameters $\theta_{i\cdot,t} = (\theta_{i1,t}, \ldots, \theta_{ik,t})$.

$\quad\quad$ • FedAvg-based: it runs $s$ steps of local gradient descent:

$$\theta_{i\cdot,t} \leftarrow \mathcal{G}_i^s(\theta_{t-1}), \quad \text{where } \mathcal{G}_i(\theta) = \theta - \eta \nabla L_i(\theta)$$

$\quad\quad$ • FedProx-based: it solves the local proximal optimization:

$$\theta_{i\cdot,t} \leftarrow \arg\min_{\theta} L_i(\theta) + \frac{1}{2\eta} \|\theta - \theta_{t-1}\|_2^2$$

$\quad$ PS updates the global model as $\theta_t \leftarrow \sum_{i=1}^{M} w_i \theta_{i\cdot,t}$, where $w_i = n_i/N$.

---

In Phase 2, the local estimation at each client has a flavor of alternating minimization: It first runs a minimization step to estimate its cluster, and then runs a FedAvg or FedProx update to refine model estimates. To allow the participation of clients with $O(1)$ data points only, at every iteration the clients are allowed to reuse all local data, including those used in the first phase. Similar alternating update is analyzed in [GCYR20] yet under the strong assumption that the update in each round is over fresh data with Gaussian distribution. Moreover, the analysis therein is restricted to the setting where the model refinement at each client is via running a single gradient step, which is barely used in practice but much simpler to analyze than FedAvg or FedProx update.

In our analysis, we consider the square loss

$$L(\theta, \lambda; \mathcal{D}_i) = \frac{1}{2n_i} \sum_{j=1}^{k} \lambda_j \|y_i - \phi(\boldsymbol{x}_i)\theta_j\|_2^2.$$

In this context, (7) yields a vertex of the probability simplex $\lambda_{ij,t} = \mathbb{1}\{j = z_{i,t}\}$, where

$$z_{i,t} = \arg\min_{j \in [k]} \|y_i - \phi(\boldsymbol{x}_i)\theta_{j,t-1}\|_2. \tag{8}$$

The estimate $z_{i,t}$ provides a hard clustering label. Hence, in each round, only one regression model will be updated per client.

To capture the tradeoff between communication cost and statistical accuracy using FedAvg or FedProx, we introduce the following quantities from [SXY21]:

$$\gamma \triangleq \eta \max_{i \in [M]} \frac{1}{n_i} \|\phi(\boldsymbol{x}_i)\|_2^2, \qquad \kappa \triangleq \begin{cases} \frac{\gamma s}{1-(1-\gamma)^s} & \text{for FedAvg,} \\ 1 + \gamma & \text{for FedProx.} \end{cases}$$

We choose a properly small learning rate $\eta$ such that $\gamma < 1$. Here, $\kappa \geq 1$ quantifies the stability of local updates. Notably, $\kappa \approx 1$ using a relatively small $\eta$.

For the learnability of model parameters, we assume that collectively there are sufficient data in each cluster. In particular, we assume $N_j \gtrsim d$, where $N_j = \sum_{i:z_i=j} n_i$ denotes the number of data points in cluster $j$. To further characterize the *quantity skewness* (i.e., the imbalance of data partition $n = (n_1, \ldots, n_M)$ across clients), we adopt the $\chi^2$-divergence, which is defined as $\chi^2(P\|Q) = \int \frac{(dP-dQ)^2}{dQ}$ for a distribution $P$ absolutely continuous with respect to a distribution $Q$. Let $\chi^2(n)$ be the chi-squared divergence between data partition $p_n$ over the clients $p_n(i) = n_i/N$ and the uniform distribution over $[M]$. Note that when data partition is balanced (i.e., $n_i = N/M$ for all $i$), it holds that $\chi^2(n) = 0$.

We have the following theoretical guarantee of Phase 2, where $s$ is the number of local steps in FedAvg. Notably, $s$ is an algorithmic parameter for FedAvg only. To recover the results for FedProx, we only need to set $s = 1$.

**Theorem 2.** *Suppose that $k \geq 2$, $\eta \lesssim 1/s$, and $N_j \gtrsim d$. Let $\rho = \min_j N_j/N$. If $\nu \log(e/\nu) \lesssim \rho/\kappa$, then with probability $1 - Cke^{-d}$, for all $t \geq T + 1$ and all $\theta_T$ such that $d(\theta_T, \theta^*) \leq \epsilon\Delta$, where $\epsilon$ is some constant, it holds that*

$$d(\theta_t, \theta^*) \leq (1 - C_1 s\eta\rho/\kappa)\, d(\theta_{t-1}, \theta^*) + C_2 s\eta\sigma\nu \log \frac{e}{\nu}, \tag{9}$$

*where*

$$\nu \triangleq \frac{1}{N} \sum_{i=1}^{M} n_i p_e(n_i) + C\sqrt{\frac{dk \log k}{M}(\chi^2(n) + 1)}, \tag{10}$$

*and $p_e(n_i) = 4ke^{-cn_i\left(1 \wedge \frac{\Delta^2}{\sigma^2}\right)^2}$. Furthermore, if $t \geq T + 1$, for each client $i$, with probability $1 - p_e(n_i)$, it is true that*

$$\left\|\hat{\theta}_{i,t} - \theta_{z_i}^*\right\|_2 \lesssim \Delta \cdot e^{-C_1 s\eta\rho(t-T)/\kappa} + \frac{\sigma\kappa}{\rho}\nu \log \frac{e}{\nu},$$

*where $\hat{\theta}_{i,t}$ is client $i$'s estimate of its own model parameter at time $t$.*

Notably, $\hat{\theta}_{i,t}$ is the $z_{i,t}$-th entry of $\theta_{i\cdot,t}$. Theorem 2 shows that the model estimation errors decay geometrically starting from any realization that is within a small neighborhood of $\theta^*$. The parameter $\nu$ captures the additional clustering errors injected at each iteration. It consists of two parts: the first term of (10) bounds the clustering error in expectation which diminishes exponentially in the local data size and the signal-to-noise ratio $\Delta/\sigma$; the second term bounds the uniform deviation of the clustering error across all initialization and iterations. Note that if the cluster structure were known exactly, we would get an model estimation error of $\theta_j^*$ scaling as $\sqrt{d/N_j}$. However, it turns out that this estimation error is dominated by our uniform deviation bound of the clustering error and hence is not explicitly shown in our bound (9). In comparison, the previous work [GCYR20] assumes fresh samples at each iteration by sample-splitting and good initialization independent of everything else provided a prior; hence their analysis fails to capture the influence of the uniform deviation of the clustering error.

In passing, we briefly comment on the key assumption $\nu \log(e/\nu) \lesssim \rho/\kappa$. As aforementioned, the clustering error $\nu$ consists of two parts shown in (10): the first term decays exponentially in the local dataset size and the signal-to-noise ratio and hence is very small in most typical scenarios; the second term is on the order of $dk \log k/M$ when the quantity skewness (imbalance of data partition) is of a constant order. Finally, $\rho$ captures imbalance of cluster sizes, which is typically of a constant order, and we can choose a small enough step size to ensure $\kappa$ is close to 1. Thus the key assumption $\nu \log(e/\nu) \lesssim \rho/\kappa$ roughly translates to $\nu$ being a subconstant, which further means that $M$ (the number of clients) needs to be larger than $d$ (the model dimension) by polylog factors. This is often satisfied in the typical FL applications which involve a very large collection of clients.

### 4.2.1 Analysis of global iterations

Without loss of generality, assume the optimal permutation in (6) is identity. In this case, if $z_{i,t} = j$, then client $i$ will refine $\theta_{j,t-1}$. To prove Theorem 2, we need to analyze the global iteration of $\theta_t$. Following a similar argument to [SXY21] with a careful examination of cluster labels, we obtain the following lemma. The proof is deferred to Appendix E.1.

**Lemma 1.** *Let $\phi(\boldsymbol{x})$ be the matrix that stacks all $\phi(\boldsymbol{x}_i)$ vertically, and similarly for y. It holds that*

$$\theta_{j,t} = \theta_{j,t-1} - \eta B \Lambda_{j,t}(\phi(\boldsymbol{x})\theta_{j,t-1} - y), \quad j \in [k], \tag{11}$$

*where $B = \frac{1}{N}\phi(\boldsymbol{x})^\top P$, $P$ is a block diagonal matrix with $i$th block $P_i$ of size $n_i \times n_i$ given by*

$$P_i = \begin{cases} \sum_{\ell=0}^{s-1}(I - \eta\phi(\boldsymbol{x}_i)\phi(\boldsymbol{x}_i)^\top/n_i)^\ell & \text{for FedAvg,} \\ [I + \eta\phi(\boldsymbol{x}_i)\phi(\boldsymbol{x}_i)^\top/n_i]^{-1} & \text{for FedProx,} \end{cases}$$

*and $\Lambda_{j,t}$ is another block diagonal matrix with $i$th block being $\lambda_{ij,t}I_{n_i}$.*

Lemma 1 immediately yields the evolution of estiamtion error. Let $\Lambda_j$ be the matrix with $i$th block being $\mathbb{1}\{z_i = j\}I_{n_i}$ representing the true client identities. Plugging model (1), the estimation error evolves as

$$\theta_{j,t} - \theta_j^* = (I - \eta K_j)(\theta_{j,t-1} - \theta_j^*) - \eta B \mathcal{E}_{j,t}(\phi(\boldsymbol{x})\theta_{j,t-1} - y) + \eta B \Lambda_j \zeta, \quad \forall j \in [k], \tag{12}$$

where $K_j = B\Lambda_j\phi(\boldsymbol{x})$ and $\mathcal{E}_{j,t} = \Lambda_{j,t} - \Lambda_j$. The estimation error is decomposed into three terms: 1) the main contribution to the decrease of estimation error; 2) the clustering error; and 3) the noisy perturbation. Let $I_j = \{i : z_i = j\}$ be the clients belonging to $j$th cluster, and $I_{j,t} = \{i : z_{i,t} = j\}$ be the clients with estimated label $j$. The indices of nonzero blocks of $\mathcal{E}_{j,t}$ are $I_j \ominus I_{j,t}$ indicating the clustering errors pertaining to $j$th cluster.

For the ease of presentation, we introduce a few additional notations for the collective data over a subset of clients. Given a subset $I \subseteq [M]$ of clients, let $\phi(\boldsymbol{x}_I)$ denote the matrix that vertically stacks $\phi(\boldsymbol{x}_i)$ for $i \in I$, and we similarly use notations $y_I$ and $\zeta_I$; let $P_I$ be the matrix with diagonal blocks $P_i$ for $i \in I$. Using those notations, we have $K_j = \frac{1}{N}\phi(\boldsymbol{x}_{I_j})^\top P_{I_j}\phi(\boldsymbol{x}_{I_j})$, which differs from the usual covariance matrix by an additional matrix $P_{I_j}$. Therefore, the analysis of the first and third terms on the right-hand side of (12) follows from standard concentration inequalities for random matrices. In the remaining of this subsection, we focus on the second term, which is a major challenge in the analysis. The proof details are all deferred to Appendix E.2.

**Lemma 2.** *There exists a universal constant $C$ such that, with probability $1 - Ce^{-d}$,*

$$\|B\mathcal{E}_{j,t}(\phi(\boldsymbol{x})\theta_{j,t-1} - y)\|_2 \lesssim s(d(\theta_{t-1}, \theta^*) + \sigma)\nu \log\frac{e}{\nu}, \quad \forall j \in [k]. \tag{13}$$

Lemma 2 aims to upper bound the error of

$$B\mathcal{E}_{j,t}(\phi(\boldsymbol{x})\theta_{j,t-1} - y) = \frac{1}{N}\phi(\boldsymbol{x}_{S_{j,t}})^\top P_{S_{j,t}}(\phi(\boldsymbol{x}_{S_{j,t}})\theta_{j,t-1} - y_{S_{j,t}}), \tag{14}$$

where $S_{j,t} = I_j \ominus I_{j,t}$. The technical difficulty arises from the involved dependency between the clustering error $S_{j,t}$ and the estimated parameter $\theta_{j,t-1}$ as estimating label $z_{i,t}$ and updating $\theta_{j,t-1}$ use a common set of local data.

*Proof Sketch of Lemma 2.* It follows from the definition of $z_{i,t}$ in (8) that

$$\|\phi(\boldsymbol{x}_i)\theta_{j,t-1} - y_i\|_2 \leq \|\phi(\boldsymbol{x}_i)\theta_{z_i,t-1} - y_i\|_2, \quad \forall i \in S_{j,t}.$$

Then,

$$\|\phi(\boldsymbol{x}_{S_{j,t}})\theta_{j,t-1} - y_{S_{j,t}}\|_2^2 = \sum_{i \in S_{j,t}} \|\phi(\boldsymbol{x}_i)\theta_{j,t-1} - y_i\|_2^2 \leq \sum_{i \in S_{j,t}} \|\phi(\boldsymbol{x}_i)\theta_{z_i,t-1} - y_i\|_2^2$$

$$\leq \sum_{i \in S_{j,t}} 2\left(\|\phi(\boldsymbol{x}_i)(\theta_{z_i,t-1} - \theta_{z_i}^*)\|_2^2 + \|\zeta_i\|_2^2\right)$$

$$\leq 2\left(d(\theta_{t-1}, \theta^*) \cdot \|\phi(\boldsymbol{x}_{S_{j,t}})\|_2 + \|\zeta_{S_{j,t}}\|_2\right)^2. \tag{15}$$

Hence, it suffices to upper bound $\|\phi(\boldsymbol{x}_{S_{j,t}})\|_2$ and $\|\zeta_{S_{j,t}}\|_2$ given a small estimation error $d(\theta_{t-1}, \theta^*)$ from the last iteration. To this end, we show a uniform upper bound of the total clustering error $\sum_{i \in S_{j,t}} n_i$ by analyzing a weighted empirical process. Using the decision rule (8), the set $S_{j,t}$ can be written as a function $S_j(\theta_{t-1})$ with

$$
\mathbb{1}\{i \in S_j(\theta)\} = \begin{cases} \max_{\ell \neq j} \mathbb{1}\{P_{j\ell}[\boldsymbol{x}_i, y_i](\theta) \geq 0\} \triangleq f_{j,\theta}^{\mathrm{I}}(\boldsymbol{x}_i, y_i), & i \in I_j, \\ \prod_{\ell \neq j} \mathbb{1}\{P_{\ell j}[\boldsymbol{x}_i, y_i](\theta) \geq 0\} \triangleq f_{j,\theta}^{\mathrm{II}}(\boldsymbol{x}_i, y_i), & i \notin I_j, \end{cases} \tag{16}
$$

where

$$
P_{jj'}[\boldsymbol{x}_i, y_i](\theta) \triangleq \|y_i - \phi(\boldsymbol{x}_i)\theta_j\|_2^2 - \|y_i - \phi(\boldsymbol{x}_i)\theta_{j'}\|_2^2.
$$

Then we derive the following uniform deviation of the incorrectly clustered data points

$$
\sup_{\theta \in \mathbb{R}^{dk}} \left| \sum_{i=1}^{M} n_i \mathbb{1}\{i \in S_j(\theta)\} - \sum_{i=1}^{M} n_i \mathbb{P}\{i \in S_j(\theta)\} \right| \leq CN \sqrt{\frac{dk \log k}{M}(\chi^2(n) + 1)}.
$$

This is proved via upper bounds on the Vapnik–Chervonenkis (VC) dimensions of the binary function classes

$$
\mathcal{F}_j^{\mathrm{I}} \triangleq \{f_{j,\theta}^{\mathrm{I}} : \theta \in \mathbb{R}^{dk}\}, \quad \mathcal{F}_j^{\mathrm{II}} \triangleq \{f_{j,\theta}^{\mathrm{II}} : \theta \in \mathbb{R}^{dk}\}. \tag{17}
$$

Using classical results of VC dimensions, those functions are equivalently intersections of hyperplanes in ambient dimension $O(d^2)$, which yields an upper bound $O(d^2)$. However, the hyperplanes are crucially rank-restricted as the total number of parameters in $\theta$ is $dk$. We prove that the VC dimensions are at most $O(dk \log k)$ using the algebraic geometry of polynomials given by the celebrated Milnor-Thom theorem (see, e.g., [Mat13, Theorem 6.2.1]).[1] Consequently, $\|\phi(\boldsymbol{x}_{S_{j,t}})\|_2$, $\|\zeta_{S_{j,t}}\|_2$ and thus (15) can be uniformly upper bounded using sub-Gaussian concentration and the union bound, concluding the proof of Lemma 2. $\qquad \square$

## 4.3 Global convergence

Combining Theorem 1 and Theorem 2, we immediately deduce the global convergence from any initialization within the $\ell_2$ ball of radius $R$.

**Theorem 3.** *Suppose the conditions of Theorem 1 and Theorem 2 hold. Let $\hat{\theta}$ be the output of our two-phase algorithm by running Phase 1 with $T = \Theta(1)$ iterations starting from any initialization $\theta_0$ with $\|\theta_0\|_2 \leq R$, followed by Phase 2 with $T' = \Theta(\frac{\kappa}{s\eta\rho} \log \frac{\Delta}{\nu})$ iterations. Then with probability $1 - N^{-9} - cke^{-d}$, it is true that*

$$
d(\hat{\theta}, \theta^*) \lesssim \frac{\sigma\kappa}{\rho} \nu \log \frac{e}{\nu}, \tag{18}
$$

*Furthermore, for each client $i$, with probability $1 - p_e(n_i)$, it holds that $\|\hat{\theta}_{i,T+T'} - \theta_{z_i}^*\|_2 \lesssim \frac{\sigma\kappa}{\rho} \nu \log \frac{e}{\nu}$.*

To the best of our knowledge, this is the first result that proves the global convergence of clustered federated learning from any initialization. Our bound (18) reveals that the final estimation error is dominated by the clustering error captured by $\nu$, and scales linearly in $\kappa$ which characterizes the stability of local updates under FedAvg or FedProx. Moreover, Theorem 3 shows that Phase 1 converges very fast with only $\Theta(1)$ iterations and hence is relatively inexpensive in both computation and communication. Instead, the number of iterations needed for Phase 2 grows logarithmically in $\Delta/\nu$ and linearly in $\kappa/(s\eta\rho)$. Thus, by choosing $s$ relatively large while keeping $\kappa$ close to 1, FedAvg enjoys a saving of the total communication cost.

## Acknowledgement

J. Xu is supported in part by the NSF Grant CCF-1856424 and an NSF CAREER award CCF-2144593. P. Yang is supported in part by the NSFC Grant 12101353 and Tsinghua University Initiative Scientific Research Program. The authors would like to thank Philippe Rigollet for pointing out the related literature on computing VC dimensions using Milnor-Thom theorem.

---

[1] Similar applications of the Milnor-Thom theorem have been known in the literature (see e.g. [GJ93, Theorem 2.2] and [VR02, Theorem 2]).

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
