\quad\quad\quad\quad\quad\quad\quad\quad\quad\quad\quad\quad\quad\quad\quad\quad\quad\quad\quad\quad\quad\quad$ */

**10** $\quad$ PS sends $\hat{U}_{i,t}$ to each anchor client $i \in H$;

**11** $\quad$ Each anchor client $i$ calls power-iteration$(A_{i,t} A_{i,t}^\top, T_2)$ to output $(\hat{\beta}_{i,t}, \hat{\sigma}_{i,t}^2)$ with $A_{i,t}$
$\quad\quad$ defined in (2);

**12** $\quad$ **if** $\hat{\sigma}_{i,t} > \epsilon\Delta$ **then**

**13** $\quad\quad$ $\theta_{i,t+1} \leftarrow \theta_{i,t} + r_{i,t}\eta_{i,t}$ and reports $\theta_{i,t+1}$, where $r_{i,t} = \hat{U}_{i,t}\hat{\beta}_{i,t}$ and
$\quad\quad\quad \eta_{i,t} = \alpha\hat{\sigma}_{i,t}/(2\beta^2)$;

**14** $\quad$ **else**

**15** $\quad\quad$ $\theta_{i,t+1} \leftarrow \theta_{i,t}$ and reports $\theta_{i,t+1}$;

**16** PS computes the pairwise distance $\|\theta_{i,T} - \theta_{i',T}\|_2$ for every pair of anchor clients $i, i' \in H$,
assigns them in the same cluster when the pairwise distance is smaller than $\Delta/2$, and
outputs $\hat{\theta}_j$ to be the center of the estimated $j$-th cluster for $j \in [k]$.

---

power-iteration and we can further approximate the leading eigenvalue of $Y$ by the Raleigh quotient $Q_t^\top Y Q_t$. When $Y$ is asymmetric, by running the orthogonal iteration on $YY^\top$, we can compute the top-$k$ left singular matrix of $Y$. In our setting, the orthogonal iteration can be implemented in a distributed manner in FL systems as shown in Algorithm 3 in the Appendix D.1.

In Step 11, each anchor client $i$ estimates the residual error $\Sigma_{z_i}(\theta_{z_i}^* - \theta_{i,t})$ by projecting $\varepsilon(x_{ij}, y_{ij}, \theta_{i,t})$ onto the previously estimated subspace, that is, $\hat{U}_{i,t}^\top \varepsilon(x_{ij}, y_{ij}, \theta_{i,t})$. This reduces the estimation from $d$-dimension to $k$-dimension and hence $\tilde{\Omega}(k)$ local data points suffice. Specifically, define

$$A_{i,t} = \frac{1}{\ell} \sum_{j \in \mathcal{D}_{i,t}} \left( \hat{U}_{i,t}^\top \varepsilon(x_{ij}, y_{ij}, \theta_{i,t}) \right) \left( \hat{U}_{i,t}^\top \varepsilon(\tilde{x}_{ij}, \tilde{y}_{ij}, \theta_{i,t}) \right)^\top, \tag{2}$$

where $\mathcal{D}_{i,t}$ consists of $2\ell$ local data points $(x_{ij}, y_{ij})$ and $(\tilde{x}_{ij}, \tilde{y}_{ij})$ freshly drawn from $\mathcal{D}_i$ at iteration $t$. Client $i$ runs the power-iteration to output $\hat{\beta}_{i,t}$ and $\hat{\sigma}_{i,t}^2$ as approximations of the leading left singular vector and singular value of $A_{i,t}$, Then anchor client $i$ updates $\theta_{i,t+1}$ by moving along the direction of the estimated residual error $r_{i,t}$ with an appropriately chosen step size $\eta_{i,t}$.

We show that $\theta_{i,T}$ is close to $\theta_{z_i}^*$ for every anchor client $i \in H$ and the outputs $\hat{\theta}_j$ are close to $\theta_j^*$ up to a permutation of cluster indices.

**Theorem 1.** *Let $\epsilon \in (0, 1/4)$ be a small but fixed constant. Suppose that*

$$m \geq p_{\min}^{-2} \tilde{\Omega}(d), \ \ell = \tilde{\Omega}(k), \ T = \Omega(1), \ T_1 = \Omega(k \log(Nd)), \ T_2 = \Omega(\log(Nd)). \tag{3}$$

*With probability at least $1 - O(n_H T/N^{10})$, for all initialization $\theta_0$ with $\|\theta_0\|_2 \leq R$,*

$$\sup_{i \in H} \left\| \theta_{i,T} - \theta_{z_i}^* \right\|_2 \leq \epsilon\Delta. \tag{4}$$

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

## A Limitations

The limitations of our work are two-fold: 1) Our setup assumes exact cluster structure, where each client belongs to a unique cluster. An interesting generalization is to consider more relaxed cluster structure, where each client may belong to a mixture of multiple clusters. 2) Our study focuses on the mixed regression model. The Phase 1 crucially relies on the regression setup. While the Phase 2 is described under the general risk minimization setup, our analysis has been restricted to mixed regression. An interesting yet challenging future direction is to extend our algorithms and analysis to more general clustered Federated Learning setting with general risk functions.

## B Broad Impact

Federated Learning has gained tremendous popularity over the past few years and has been widely deployed in products such as Apple's Siri and Google's Gboard. It opens up a world of new opportunities for training machine learning models without compromising data privacy. Our paper significantly advances the theory and algorithms for Federated Learning, providing important design insights and key enabling technologies for efficient and personalized model training under FL. Our study is highly interdisciplinary, bringing together ideas from statistics, optimization, and distributed computation. We are unaware of any potential negative societal impacts of our work.

## C Experimental results

In this section, we provide experimental results on synthetic data corroborating our theoretical findings.

We consider the mixed linear regression with $k = 3$ clusters. The true model parameter for each cluster $\theta_1^*, \theta_2^*, \theta_3^*$ are independently sampled from Gaussian distribution $\frac{2}{\sqrt{d}} * \mathcal{N}(0, I_d)$ with $d = 100$. Then we generate the local dataset $\mathcal{D}_i = \{x_{ij}, y_{ij}\}_{j=1}^{n_i}$ for each client $i$ according to the linear regression model (1), where each $x_{ij} \overset{\text{i.i.d.}}{\sim} \mathcal{N}(0, I_d)$ and $\zeta_{ij} \overset{\text{i.i.d.}}{\sim} 0.2 * \mathcal{N}(0, 1)$.

We simulate our two-phase algorithm as follows. Phase 1 randomly selects $\lceil 3k \log k \rceil$ anchors clients and runs 5 iterations starting from a random initialization $\frac{2}{\sqrt{d}} * \mathcal{N}(0, I_d)$, followed by Phase 2 running 400 global iterations. We further adopt the following simplifications for ease of implementation. In particular, Phase 1 reuses the local data on all participating clients, and all clients including anchor clients participate in the subspace estimation subroutine in Algorithm 3. Finally, we implement all orthogonal iterations by direct singular value decomposition.

We compare the performance of our two-phase algorithm with existing FL algorithms including (1) vanilla FedAvg, (2) one-shot clustering, (3) IFCA, and (4) oracle iterative clustering.

(1) The vanilla FedAvg ignores the underlying cluster structure and learns a common model.

(2) In the one-shot clustering [GHYR19], first each client estimates its underlying model based on its local data, then the PS clusters the locally estimated models via $k$-means, and finally within each estimated cluster, we run FedAvg to obtain the model estimate for each cluster.

(3) The IFCA algorithm [GCYR20] is the same as Phase 2 of our algorithm.

(4) Oracle iterative clustering algorithm is an ideal implementation of IFCA initialized with the true model parameters. Clearly, the oracle iterative clustering algorithm is infeasible in practice, but we use it as a benchmark.

For each of the methods, we choose FedAvg with the number of local update steps $s = 5$. We randomly initialize our two-phase algorithm, vanilla FedAvg, and IFCA.

In the following, we consider three federated learning settings with a total of $N = 1000$ data points but at increasing levels of data heterogeneity.

### C.1 Balanced local data and balanced cluster partition

In this configuration, we consider balanced local data and balanced cluster partition. Specifically, we let $M = 200$, $n_i = 50$ for $i \in [M]$, and $p_1 = p_2 = p_3 = 1/3$. That is, this configuration contains

200 clients, each with 50 data points. For each client, it belongs to one of the 3 clusters with equal probability $1/3$.

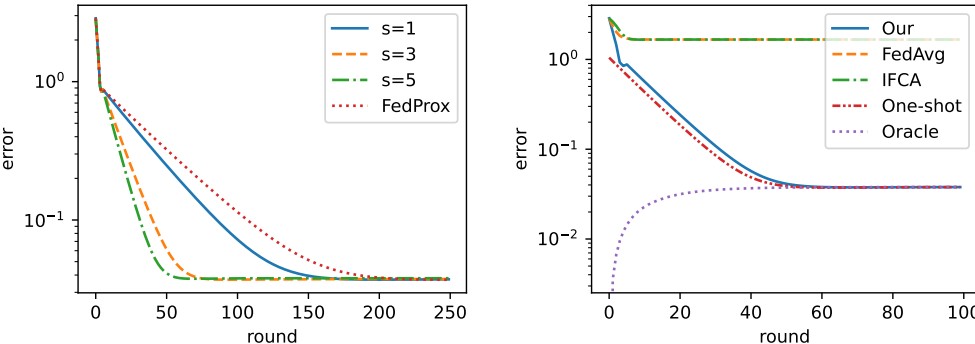

**Figure 1:** Balanced local data and balanced cluster partition.

In the left panel of Fig. 1, we show the performance of our two-phase algorithm, where the second phase is based on FedAvg for different local steps $s$ or FedProx. We see that during the first 5 rounds (Phase 1), the errors quickly (exponentially with a large rate) converge to a relatively small value. Starting from iteration 6 (upon entering Phase 2), the errors further decay exponentially fast (with a smaller rate than Phase 1). These observations are consistent with our theoretical predictions. We also notice that as the number of local steps $s$ increases, FedAvg converges faster, while the final estimation errors stay almost the same. This is because the data partition is perfectly balanced, so the local updates of FedAvg are relatively stable with $\kappa \approx 1$; hence according to Theorem 2 and Theorem 3, the convergence rate increases proportionally to $s$, while the final estimation does not change.

The right panel of Fig. 1 shows that our method significantly outperforms vanilla FedAvg and IFCA, and quickly converges to the same estimation error attainable by the oracle algorithm. Note that FedAvg does not converge to small errors due to lack of model personalization in the presence of model heterogeneity. The performance of IFCA is highly dependent on the quality of initialization. With a random initialization, IFCA gets stuck on an error floor. The one-shot clustering algorithm performs well in this setting. This is because the local data partition and cluster partition are perfectly balanced, so each client can well estimate its underlying model solely based on its local data and the PS can correctly cluster all the locally estimated models via $k$-means.

## C.2  Unbalanced local data and balanced cluster partition

In this configuration, we consider unbalanced local data but balanced cluster partition. Specifically, we let $M = 920$, $n_i = 10$ for $i = 1, \cdots, 900$, and $n_i = 50$ for $i = 901, \cdots, 920$. That is, this configuration contains 920 clients, with each of the first 900 clients keeps 10 data points, and each of the remaining 20 clients keeps 50 data points. For each client, it belongs to one of the 3 clusters with equal probability $1/3$.

The left panel of Fig. 2 stays almost the same as that of Fig. 1. The only noticeable difference is that in this setting with unbalanced local data, as $s$ increases, the convergence rate of FedAvg only slightly improves, while the final estimation error also gets slightly inflated. This is because with unbalanced local data, the local updates of FedAvg for data-scarce clients become unstable, leading to a larger value of $\kappa$.

The right panel of Fig. 2 shows that our method still significantly outperforms vanilla FedAvg and IFCA, and quickly converges to the same estimation error attainable by the oracle algorithm. Although this time IFCA eventually also converges to the oracle estimation error, it still gets stuck on an error floor for a long time. The one-shot clustering algorithm no longer performs as well as before. This is because here the local data partition is unbalanced, so data-scarce clients cannot well estimate their underlying models solely based on their local data and the PS is likely to incorrectly cluster them. Since in one-shot clustering, the clustering is done only once and fixed throughout the remaining process, these clustering errors cannot be corrected.

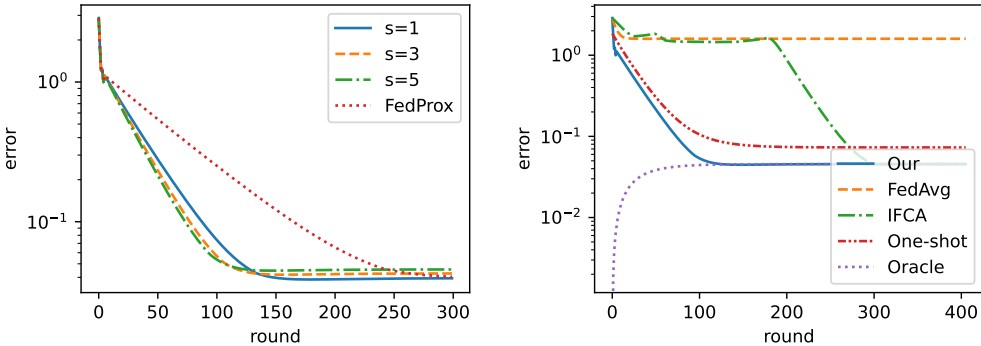

**Figure 2:** Unbalanced local data and balanced cluster partition.

### C.3 Unbalanced local data and unbalanced cluster partition

In this configuration, we consider unbalanced local data and unbalanced cluster partition. Specifically, we let $M = 920$, $n_i = 10$ for $i = 1, \cdots, 900$, and $n_i = 50$ for $i = 901, \cdots, 920$. That is, this configuration contains 920 clients, with each of the first 900 clients keeps 10 data points, and each of the remaining 20 clients keeps 50 data points. For each client, it belongs to one of the 3 clusters with probability $p_1 = 0.2$, $p_2 = 0.3$, $p_3 = 0.5$.

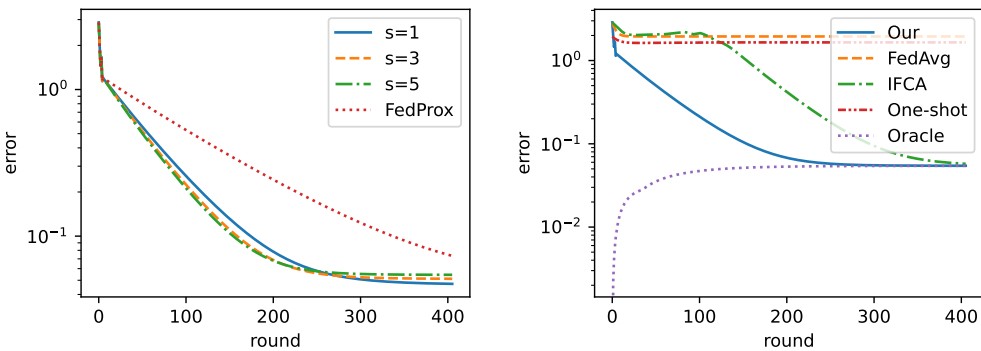

**Figure 3:** Unbalanced local data and unbalanced cluster partition.

The left panel of Fig. 3 stays almost the same as that of Fig. 2, except that convergence rates get smaller. This is consistent with our theoretical prediction in Theorem 2, which shows that the convergence rate is proportional to $\rho$ (roughly the same as $p_{\min}$).

The right panel of Fig. 3 shows that our method significantly outperforms vanilla FedAvg, IFCA, and one-shot, and quickly converges to the same estimation error attainable by the oracle algorithm. Note that one-shot clustering performs poorly in this case, because with unbalanced local data partition and unbalanced cluster partition, the one-shot clustering suffers from a large amount of errors in the initial clustering based on locally estimated models.

## D Analysis of Phase 1

In this section, we present the analysis of our federated moment descent algorithm as described in Phase 1.

### D.1 Subspace estimation via federated orthogonal iteration

Recall that Phase 1 aims to estimate the subspace that the residual estimation errors $\{\Sigma_j(\theta_j^* - \theta_{i,t})\}_{j=1}^k$ lie in via the federated-orthogonal iteration. We can show that $\mathbb{E}[Y_{i,t}]$ is of rank at most $k$

and the eigenspace corresponding to the non-zero eigenvalues is spanned by $\{\Sigma_j(\theta_j^* - \theta_{i,t})\}_{j=1}^k$. Specifically, we first prove that $Y_{i,t}$ is close to $\mathbb{E}[Y_{i,t}]$ in the operator norm and then further deduce that $\{\Sigma_j(\theta_j^* - \theta_{i,t})\}_{j=1}^k$ approximately lie in the subspace spanned by the top-$k$ left singular vectors of $Y_{i,t}$.

Let $U_{i,t} \in \mathbb{R}^{d \times k}$ denote the top-$k$ left singular matrix of $Y_{i,t}$. To approximately compute $U_{i,t}$ in the FL systems, we adopt the following **federated-orthogonal iteration** algorithm. Suppose that $Y$ admits a decomposition over distributed clients, that is, $Y = \frac{1}{\sum_{i \in S} n_i} \sum_{i \in S} \sum_{j \in n_i} a_{ij} b_{ij}^\top$, where $S$ is a set of clients, and $\{(a_{ij}, b_{ij})\}_{j=1}^{n_i}$ are computable based on the local dataset $\mathcal{D}_i$. Algorithm 3 approximates the top-$k$ left singular matrix of $Y$. It can be easily verified that Algorithm 3 effectively runs the orthogonal iteration on $YY^\top$.

---

**Algorithm 3:** Federated orthogonal iteration

---

1  **Input:** A set $\mathcal{S}$ of clients $i$ with $\{(a_{ij}, b_{ij})\}_{i \in \mathcal{S}, j \in [n_i]}$, $k \in \mathbb{N}$, and even $T \in \mathbb{N}$

2  **Output:** $Q_T \in \mathbb{R}^{d \times k}$

  1: PS initializes $Q_0 \in \mathbb{R}^{d \times k}$ as a random orthogonal matrix $Q_0^\top Q_0 = \mathbf{I}$.

  2: **for** $t = 0, 1, \ldots, T - 1$ **do**

  3:      PS broadcasts $Q_t$ to all clients in $\mathcal{S}$.

  4:      **if** $t$ is even **then**

  5:         Each client $i \in S$ computes an update $Q_{i,t} = \frac{1}{n_i} \sum_{j=1}^{n_i} b_{ij} a_{ij}^\top Q_t$ and transmits it back to the PS.

  6:         PS updates $Q_{t+1} = \sum_{i \in S} w_i Q_{i,t}$, where $w_i = n_i / \sum_{i \in S} n_i$.

  7:      **else**

  8:         Each client in $\mathcal{S}$ computes an update $Q_{i,t} = \frac{1}{n_i} \sum_{j=1}^{n_i} a_{ij} b_{ij}^\top Q_t$ and transmits it back to the PS.

  9:         PS applies the QR decomposition to obtain $Q_{t+1}$:

$$\sum_{i \in \mathcal{S}} w_i Q_{i,t} = Q_{t+1} R_{t+1}.$$

  10:    **end if**

  11: **end for**

---

Recall that Phase 1 is called for each anchor client $i \in H$ and each global iteration $t$, and that $\hat{U}_{i,t}$ is the output of **federated-orthogonal iteration** in Step 9 of Phase 1, which approximates $U_{i,t}$. Based on the above discussion, we can show that the residual estimation errors $\{\Sigma_j(\theta_j^* - \theta_{i,t})\}_{j=1}^k$ approximately lie in the subspace spanned by the $k$ columns of $\hat{U}_{i,t}$.

**Proposition 1** (Subspace estimation). *If $T_1 \geq Ck \log(Nd)$ for some sufficiently large constant $C$, then with probability at least $1 - N^{-10}$,*

$$\left\| \left( \hat{U}_{i,t} \hat{U}_{i,t}^\top - I \right) \Sigma_j \left( \theta_j^* - \theta_{i,t} \right) \right\|_2^2 \leq O\left( \left( \delta_{i,t}^2 + \sigma^2 \right) \xi_1 / p_j \right), \quad \forall j \in [k],$$

*where $\delta_{i,t} = \max_{j \in [k]} \|\theta_j^* - \theta_{i,t}\|_2$ and $\xi_1 = \sqrt{\frac{d}{m} \log N} + \frac{d}{m} \log^3 N$.*

We postpone the detailed proof to Appendix D.4. One key challenge in the analysis is that the eigengap of $\mathbb{E}[Y_{i,t}]$ could be small, especially when $\theta_{i,t}$ is close to $\theta_{z_i}^*$; and hence the standard Davis-Kahan theorem cannot be applied. This issue is further exaggerated by the fact that the convergence rate of the orthogonal iteration also crucially depends on the eigengaps. To resolve this issue, one key innovation of our analysis is to develop a gap-free bound to show that the projection errors $\hat{U}_{i,t}^\top \Sigma_j (\theta_j^* - \theta_{i,t})$ are small for every $j \in [k]$ (cf. Lemma 5).

## D.2   Moment descent on anchor clients

Recall that in Step 11 of Phase 1, each anchor client $i \in H$ runs the **power-iteration** to output $\hat{\beta}_{i,t}$ and $\hat{\sigma}_{i,t}^2$ as approximations of the leading left singular vector and singular value of $A_{i,t}$, respectively.

Then anchor client $i$ updates a new estimate $\theta_{i,t+1}$ by moving along the direction of the estimated residual error $r_{i,t}$ with an appropriately and adaptively chosen step size $\eta_{i,t}$. The following result shows that $\hat{\sigma}_{i,t}^2$ closely approximates the squared residual error $\|\Sigma_j(\theta_{z_i}^* - \theta_{i,t})\|_2^2$. Moreover, the residual error decreases geometrically until reaching a plateau.

**Proposition 2.** *Fix an anchor client $i$ and suppose $T_2 \gtrsim \log(Nd)$. There exists a constant $C > 0$ and an event $\mathcal{E}_{i,t}$ with $\mathbb{P}\{\mathcal{E}_{i,t}\} \geq 1 - O(N^{-10})$ such that on event $\mathcal{E}_{i,t}$*

$$\left| \left\| \Sigma_j (\theta_{z_i}^* - \theta_{i,t}) \right\|_2^2 - \hat{\sigma}_{i,t}^2 \right| \leq C(\delta_{i,t}^2 + \sigma^2)(\xi_1/p_{z_i} + \xi_2), \tag{19}$$

*where $\xi_2 = \sqrt{\frac{k}{\ell} \log N} + \frac{k}{\ell} \log^3 N$. Furthermore, if*

$$\left\| \Sigma_{z_i} (\theta_{z_i}^* - \theta_{i,t}) \right\|_2^2 \geq C(\xi_1/p_{z_i} + \xi_2), \tag{20}$$

*then*

$$\left\| \Sigma_{z_i} (\theta_{z_i}^* - \theta_{i,t+1}) \right\|_2^2 \leq \left(1 - \frac{\alpha^2}{8\beta^2}\right) \left\| \Sigma_{z_i} (\theta_{z_i}^* - \theta_{i,t}) \right\|_2^2, \tag{21}$$

*We postpone the proof to Appendix D.5. The key ingredient in the proof of geometric decay (21) is to show that the descent direction $r_{i,t}$ is approximately parallel to the residual error $\Sigma_{z_i}(\theta_{z_i}^* - \theta_{i,t})$ under condition (20)*

### D.3 Proof of Theorem 1

Now, we are ready to prove our main theorem on the performance guarantee of Phase 1. Let $\mathcal{E}_{i,t}$ denote the event under which the statement of Proposition 2 holds. Let $\mathcal{E} = \cup_{i \in H} \cup_{t=1}^T \mathcal{E}_{i,t}$. By the union bound, $\mathbb{P}\{\mathcal{E}\} \geq 1 - O(n_H T/N^{10})$. In the following, we assume event $\mathcal{E}$ holds.

We first prove (4). Fix any anchor client $i$ and omit the subscript $i$ for simplicity. We further assume it belongs to cluster $j$, i.e., $z_i = j$. Define

$$t^* = \min\{\inf\{t \geq 0 : \hat{\sigma}_t \leq \epsilon\Delta\}, T\}.$$

By definition,

$$\hat{\sigma}_t > \epsilon\Delta, \quad \forall 0 \leq t \leq t^* - 1. \tag{22}$$

Moreover, by the update rule of our algorithm, $\theta_T = \theta_{t^*}$. Thus, it suffices to bound $\|\theta_{t^*} - \theta_j^*\|_2$.

We claim that for all $0 \leq t \leq t^*$,

$$\left\| \Sigma_j (\theta_j^* - \theta_t) \right\|_2^2 \leq \left(1 - \frac{\alpha^2}{8\beta^2}\right)^t \left\| \Sigma_j (\theta_j^* - \theta_0) \right\|_2^2 \tag{23}$$

$$\|\theta_t\|_2 \leq (1 + 2\beta/\alpha)R. \tag{24}$$

If $t^* = T$, then by (23) we immediately get that

$$\left\| \Sigma_j (\theta_j^* - \theta_{t^*}) \right\|_2 \leq \left(1 - \frac{\alpha^2}{8\beta^2}\right)^{T/2} \left\| \Sigma_j \theta_j^* \right\|_2 \leq \exp\left(-T\alpha^2/(16\beta^2)\right) \beta R \leq \epsilon\Delta,$$

where the last inequality holds by choosing $T = \frac{16\beta^2}{\alpha^2} \log \frac{\beta R}{\epsilon\Delta}$.

If $t^* < T$, then by (24) and $\|\theta_j^*\|_2 \leq R$ for all $j \in [k]$, it follows that $\delta_t \leq 2(1 + \beta/\alpha)R$. Therefore, by (19)

$$\left| \left\| \Sigma_j (\theta_j^* - \theta_t) \right\|_2^2 - \hat{\sigma}_t^2 \right| \leq C(\delta_t^2 + \sigma^2)(\xi_1/p_j + \xi_2) \leq \epsilon^2\Delta^2/2, \tag{25}$$

where the last inequality holds by choosing $m = \tilde{\Omega}(d/p_{\min}^2)$ and $\ell = \tilde{\Omega}(k)$ and invoking the standing assumption that $R = O(\Delta)$ and $\sigma = O(\Delta)$. It immediately follows that

$$\left\| \Sigma_j (\theta_j^* - \theta_{t^*}) \right\|_2^2 \leq \hat{\sigma}_{t^*}^2 + \epsilon^2\Delta^2/2 \leq \frac{3}{2}\epsilon^2\Delta^2,$$

where the last inequality holds by the stopping rule of our algorithm so that $\hat{\sigma}_{t^*} \leq \epsilon\Delta$.

In both cases, we get that
$$\left\|\theta_j^* - \theta_{t^*}\right\|_2 \le \frac{1}{\alpha} \left\|\Sigma_j(\theta_j^* - \theta_{t^*})\right\|_2 \le \frac{2\epsilon\Delta}{\alpha} = \epsilon'\Delta$$
for $\epsilon' = 2\epsilon/\alpha$. This proves (4).

Now, it remains to prove the claim (23)–(24) by induction. The base case $t = 0$ trivially holds as $\|\theta_0\|_2 \le R$. Now, suppose the induction hypothesis holds for an arbitrary $t$ where $0 \le t \le t^* - 1$, we prove it also holds for $t + 1$. In view of (25),
$$\left\|\Sigma_j(\theta_j^* - \theta_t)\right\|_2^2 \ge \hat{\sigma}_t^2 - \epsilon^2\Delta^2/2 > \epsilon^2\Delta^2/2 \ge C(\delta_t^2 + \sigma^2)(\xi_1/p_j + \xi_2),$$
where the second inequality holds due to (22). Therefore, the condition (20) is satisfied. Hence, by applying Proposition 2, we get that
$$\left\|\Sigma_j\left(\theta_j^* - \theta_{t+1}\right)\right\|_2^2 \le \left(1 - \frac{\alpha^2}{8\beta^2}\right) \left\|\Sigma_j\left(\theta_j^* - \theta_t\right)\right\|_2^2 \le \left(1 - \frac{\alpha^2}{8\beta^2}\right)^{t+1} \left\|\Sigma_j\left(\theta_j^* - \theta_0\right)\right\|_2^2.$$
where the second inequality holds by the induction hypothesis (23). Hence
$$\alpha^2 \left\|\theta_j^* - \theta_{t+1}\right\|_2^2 \le \left\|\Sigma_j\left(\theta_j^* - \theta_{t+1}\right)\right\|_2^2 \le \left\|\Sigma_j\left(\theta_j^* - \theta_0\right)\right\|_2^2 \le 4\beta^2 R^2.$$
It follows that
$$\left\|\theta_j^* - \theta_{t+1}\right\|_2 \le \frac{2\beta}{\alpha}R$$
and hence $\|\theta_{t+1}\|_2 \le (1 + 2\beta/\alpha)R$. This completes the induction proof.

Finally, we prove (5). Note that by standard coupon collector's problem, we deduce that if $n_H \ge \log(k/\delta)/p_{\min}$, then with probability at least $1 - \delta$, $H \cap \{i : z_i = j\} \ne \emptyset$ for all $j \in [k]$. To see this, note that
$$\mathbb{P}\{H \cap \{i : z_i = j\} \ne \emptyset, \forall j \in [k]\} \ge 1 - \sum_{j \in [k]} \mathbb{P}\{H \cap \{i : z_i = j\} = \emptyset\}$$
$$\ge 1 - k\left(1 - p_{\min}\right)^{n_H}$$
$$\ge 1 - k\exp\left(-p_{\min}n_H\right) \ge 1 - \delta.$$
Therefore, as long as $\epsilon' < 1/4$, we have for two anchor clients $i, i' \in H$
$$\|\theta_{i,T} - \theta_{i',T}\|_2 \le \|\theta_{i,T} - \theta_{z_i}^*\|_2 + \|\theta_{i',T} - \theta_{z_{i'}}^*\|_2 \le 2\epsilon'\Delta, \quad \text{if } z_i = z_{i'}$$
$$\|\theta_{i,T} - \theta_{i',T}\|_2 \ge \Delta - \|\theta_{i,T} - \theta_{z_i}^*\|_2 - \|\theta_{i',T} - \theta_{z_{i'}}^*\|_2 \ge (1 - 2\epsilon')\Delta, \quad \text{if } z_i \ne z_{i'}.$$
Thus, by assigning anchor clients $i, i' \in H$ in the same cluster when $\|\theta_{i,T} - \theta_{i',T}\|_2 \le \Delta/2$ we can exactly recover the $k$ clusters of the clients users. In particular, let $\hat{z}_i$ denote the estimated cluster label of anchor client $i \in H$. Then there exists a permutation $\pi : [k] \to [k]$ such that $\pi(\hat{z}_i) = z_i$ for all $i \in H$. Let $\hat{\theta}_j$ denote the center of the recovered cluster $j$, that is
$$\hat{\theta}_j = \sum_{i \in H} \theta_{i,T}\mathbb{1}\{\hat{z}_i = j\}/\sum_{i \in H}\mathbb{1}\{\hat{z}_i = j\}.$$
Then we have $\|\hat{\theta}_{\pi(j)} - \theta_j^*\|_2 \le \epsilon'\Delta$ for all $j \in [k]$. This finishes the proof of (5).

### D.4 Proof of Proposition 1

In the following analysis, we fix an anchor client $i \in H$ and omit the subscript $i$ for ease of presentation. Crucially, since $\mathcal{S}_t$ and $\mathcal{D}_t$ are freshly drawn, all the global data and local data used in iteration $t + 1$ are independent from $\theta_t$. Hence, we condition on $\theta_t$ and $\mathcal{S}_t$ in the following analysis. Note that
$$\mathbb{E}[Y_t] = \frac{1}{m}\sum_{i'\mathcal{S}_t}\mathbb{E}_{z_{i'}}\left[\Sigma_{z_{i'}}\left(\theta_{z_{i'}}^* - \theta_t\right)\left(\theta_{z_{i'}}^* - \theta_t\right)\Sigma_{z_{i'}}\right] = \sum_{j=1}^k p_j\Sigma_j\left(\theta_j^* - \theta_t\right)\left(\theta_j^* - \theta_t\right)^\top\Sigma_j,$$
where $p_j$ is the probability that a client belongs to the $j$-th cluster.

Let $U_t \in \mathbb{R}^{d \times k}$ denote the left singular matrix of $Y_t$. We aim to show that the collection of $\Sigma_j(\theta_j^* - \theta_t)$ for $j \in [k]$ approximately lie in the space spanned by the $k$ columns of $U_t$. As such, we first show that $Y_t$ is close to $\mathbb{E}[Y_t]$ in operator norm.

**Lemma 3.** *With probability at least* $1 - 3N^{-10}$,

$$\|Y_t - \mathbb{E}[Y_t]\|_2 \leq O\left(\left(\delta_t^2 + \sigma^2\right)\xi_1\right),$$

*where* $\delta_t = \max_{j \in [k]} \|\theta_j^* - \theta_t\|_2$ *and* $\xi_1 = \sqrt{\frac{d}{m}\log N} + \frac{d}{m}\log^3 N$.

*Proof.* Let $\varepsilon_i = (y_{i1} - \langle \phi(x_{i1}), \theta_t \rangle)\phi(x_{i1})$ and $\tilde{\varepsilon}_i = (y_{i2} - \langle \phi(x_{i2}), \theta_t \rangle)\phi(x_{i2})$. Note that

$$Y_t - \mathbb{E}[Y_t] = \frac{1}{m}\sum_{i=1}^m \varepsilon_i \tilde{\varepsilon}_i^\top - \mathbb{E}\left[\varepsilon_i \tilde{\varepsilon}_i^\top\right].$$

Let $a_i = \varepsilon_i/\sqrt{\delta_t^2 + \sigma^2}$ and $b_i = \tilde{\varepsilon}_i/\sqrt{\delta_t^2 + \sigma^2}$. We will apply a truncated version of the Matrix Bernstein's inequality given in Lemma 13. As such, we first check the conditions in Lemma 13 are all satisfied. Note that

$$\mathbb{E}\left[\|\varepsilon_i\|_2^2\right] = \mathbb{E}\left[\left\|\left(\langle\phi(x_{i1}), \theta_{z_i}^* - \theta_t\rangle + \zeta_i\right)\phi(x_{i1})\right\|_2^2\right]$$
$$= \mathbb{E}\left[\left\|\langle\phi(x_{i1}), \theta_{z_i}^* - \theta_t\rangle\phi(x_{i1})\right\|_2^2\right] + \mathbb{E}\left[\|\zeta_{i1}\phi(x_{i1})\|_2^2\right].$$

By the sub-Gaussianity of $\phi(x_{i1})$, we have

$$\mathbb{E}\left[\|\zeta_i\phi(x_{i1})\|_2^2\right] \leq \sigma^2 \mathbb{E}\left[\|\phi(x_{i1})\|_2^2\right] = O(\sigma^2 d)$$

and further by Cauchy-Schwarz inequality,

$$\mathbb{E}\left[\left\|\langle\phi(x_{i1}), \theta_{z_i}^* - \theta_t\rangle\phi(x_{i1})\right\|_2^2\right] \leq \sqrt{\mathbb{E}\left[\langle\phi(x_{i1}), \theta_{z_i}^* - \theta_t\rangle^4\right]}\sqrt{\mathbb{E}\left[\|\phi(x_{i1})\|_2^4\right]} = O\left(\delta_t^2 d\right).$$

Combining the last three displayed equation gives that $\mathbb{E}\left[\|a_i\|_2^2\right] \leq O(d)$. The same upper bound also holds for $\mathbb{E}\left[\|b_i\|_2^2\right]$.

Moreover, $\left\|\mathbb{E}\left[a_i a_i^\top\right]\right\|_2 = \sup_{u \in \mathcal{S}^{d-1}} \mathbb{E}\left[\langle a_i, u\rangle^2\right]$. Note that for any $u \in \mathcal{S}^{d-1}$,

$$\mathbb{E}\left[\langle a_i, u\rangle^2\right] = \frac{1}{\delta_t^2 + \sigma^2}\mathbb{E}\left[r_i^2\langle\phi(x_{i1}), u\rangle^2\right] \leq \frac{1}{\delta_t^2 + \sigma^2}\sqrt{\mathbb{E}[r_i^4]}\sqrt{\langle\phi(x_{i1}), u\rangle^4} \leq O(1),$$

where $r_i = y_{i1} - \langle\phi(x_{i1}), \theta_t\rangle$. Combining the last two displayed equations gives that $\left\|\mathbb{E}\left[a_i a_i^\top\right]\right\|_2 = O(1)$. The same upper bound also holds for $\left\|\mathbb{E}\left[b_i b_i^\top\right]\right\|_2$. Finally, by the sub-Gaussian property of $\phi(x_{i1})$, we have

$$\mathbb{P}\{\|\phi(x_{i1})\|_2 \geq s_1\} \leq \exp\left(O(d) - \Omega(s_1^2)\right)$$

and

$$\mathbb{P}\left\{\frac{|r_i|}{\sqrt{\delta_t^2 + \sigma^2}} \geq s_2\right\} \leq \exp\left(-\Omega\left(s_2^2\right)\right).$$

Choosing $s_1 = C\sqrt{s}d^{1/4}$ and $s_2 = \sqrt{s}/(Cd^{1/4})$ for a sufficiently large constant $C$, we get that for all $s \geq \sqrt{d}$,

$$\mathbb{P}\{\|a_i\|_2 \geq s\} \leq \mathbb{P}\{\|\phi(x_{i1})\|_2 \geq s_1\} + \mathbb{P}\left\{\frac{|r_i|}{\sqrt{\delta_t^2 + \sigma^2}} \geq s_2\right\}$$
$$\leq \exp\left(O(d) - \Omega(Cs\sqrt{d})\right) + \exp\left(-\Omega\left(\frac{s}{\sqrt{d}}\right)\right)$$
$$\leq \exp\left(-\Omega\left(\frac{s}{\sqrt{d}}\right)\right).$$

The same bound holds for $\mathbb{P}\{\|b_i\|_2 \geq s\}$. Applying the truncated version of the Matrix Bernstein's inequality given in Lemma 13 yields the desired result. $\square$

The following result shows the geometric convergence of orthogonal iteration. Let $Y = U\Lambda U^\top$ denote the eigenvalue decomposition of $Y$ with $|\lambda_1| \geq |\lambda_2| \geq \cdots |\lambda_d|$ and the corresponding eigenvectors $u_i$'s. Define $U_1 = [u_1, \ldots, u_k]$ and $U_2 = [u_{k+1}, \ldots, u_d]$.

**Lemma 4.** *[GVL13, Theorem 8.2.2] Assume $|\lambda_k| > |\lambda_{k+1}|$ and $\cos(\gamma) = \sigma_{\min}(U_1^\top Q_0)$ for $\gamma \in [0, \pi/2]$. Then*

$$\left\| Q_t Q_t^\top - U_1 U_1^\top \right\|_2 \leq \tanh(\theta) \left| \frac{\lambda_{k+1}}{\lambda_k} \right|^t, \quad \forall t.$$

Finally, we need a gap-free bound that controls the projection errors.

**Lemma 5** (Gap-free bound on projection errors). *Suppose $M \in \mathbb{R}^{d \times d}$ satisfies that*

$$\left\| M - \sum_{i=1}^k x_i x_i^\top \right\|_2 \leq \epsilon,$$

*where $x_i \in \mathbb{R}^d$ for $1 \leq i \leq k$. Let $Q_t$ be the output of the orthogonal iteration running over $MM^\top$. Assume that $\|x_i\|_2 \leq H$ for all $1 \leq i \leq k$. There exists a universal constant $C > 0$ such that for any $\epsilon > 0$ and $t \geq Ck \log \frac{dNH}{\epsilon}$, we have with probability at least $1 - O(N^{-10})$,*

$$\left\| Q_t Q_t^\top x_i - x_i \right\|_2 \leq 3\sqrt{\epsilon}, \quad \forall 1 \leq i \leq k.$$

*Proof.* Let $\sigma_1 \geq \sigma_2 \geq \ldots \geq \sigma_d \geq 0$ denote the singular values of $M$. Then by assumption on $M$ and Weyl's inequality, $\sigma_{k+1} \leq \epsilon$. We divide the analysis into two cases depending on the value of $\sigma_1$. Let $\delta > 0$ be some parameter to be tuned later.

**Case 1**: $\sigma_1 \leq (1+\delta)^k \epsilon$. In this case, by Weyl's inequality,

$$\|x_i\|_2^2 \leq \left\| \sum_{i=1}^k x_i x_i^\top \right\|_2 \leq \|M\|_2 + \left\| M - \sum_{i=1}^k x_i x_i^\top \right\|_2 \leq \epsilon \left( 1 + (1+\delta)^k \right).$$

Thus,

$$\left\| Q_t Q_t^\top x_i - x_i \right\|_2 \leq \|x_i\|_2 \leq \sqrt{\epsilon \left( 1 + (1+\delta)^k \right)}$$

**Case 2**: $\sigma_1 > (1+\delta)^k \epsilon$. Then by the pigeonhole principle there must exist $1 \leq p \leq k$ such that $\sigma_p/\sigma_{p+1} > 1 + \delta$. Choose

$$\ell = \max \{p : \sigma_p/\sigma_{p+1} > 1 + \delta\}.$$

It follows that $\sigma_{\ell+1} \leq (1+\delta)^{k-\ell}\epsilon \leq (1+\delta)^k \epsilon$. Let $U_\ell = [u_1, \ldots, u_\ell]$, where $u_i$'s are the left singular vectors of $M$ corresponding to $\sigma_i$. Given the subspace $\text{span}\{u_1, \ldots, u_\ell\}$, denote the unique orthogonal decomposition of $x_i$ by $x_i = \Pi_W(x_i) + e$, where $\Pi_W(x_i) = U_\ell U_\ell^\top x_i$ and $e^\top u_j = 0$ for all $j \in [\ell]$. Let $u = e/\|e\|_2 \in S^{d-1}$. Then,

$$\left\| U_\ell U_\ell^\top x_i - x_i \right\|_2^2 = u^\top x_i x_i^\top u \leq u^\top \left( \sum_{i=1}^k x_i x_i^\top \right) u = u^\top \left( \sum_{i=1}^k x_i x_i^\top - M \right) u + u^\top M u.$$

Note that

$$u^\top \left( \sum_{i=1}^k x_i x_i^\top - M \right) u \leq \left\| \sum_{i=1}^k x_i x_i^\top - M \right\|_2 \leq \epsilon.$$

Moreover,

$$u^\top M u = \sum_j \sigma_j u^\top u_j v_j^\top u = \sum_{j \geq \ell+1} \sigma_j u^\top u_j v_j^\top u \leq \sigma_{\ell+1} \sum_{j \geq \ell+1} |u^\top u_j| |v_j^\top u|$$

$$\leq \sigma_{\ell+1} \sqrt{\sum_{j \geq \ell+1} |u^\top u_j|^2 \sum_{j \geq \ell+1} |v_j^\top u|^2}$$

$$\leq \sigma_{\ell+1} \leq (1+\delta)^k \epsilon.$$

Combining the last three displayed equations gives that

$$\left\|U_\ell U_\ell^\top x_i - x_i\right\|_2^2 \leq \epsilon\left(1 + (1+\delta)^k\right).$$

Let $\hat{Q}_t$ be the submatrix of $Q_t$ formed by the first $\ell$ columns. Since $\sigma_\ell > \sigma_{\ell+1}$, the space spanned by $\hat{Q}_t$ is the same space spanned by $Q_t$ if the orthogonal iteration were run with $k$ replaced by $\ell$. Thus, applying Lemma 4 with $k$ replaced by $\ell$ gives that

$$\left\|\hat{Q}_t\hat{Q}_t^\top - U_\ell U_\ell^\top\right\|_2 \leq \tan(\gamma)(1+\delta)^{-t},$$

where $\cos(\gamma) = \sigma_{\min}(U_\ell^\top \hat{Q}_0)$ and $\hat{Q}_0$ is the submatrix of $Q_0$ formed by its first $\ell$ columns. Applying Lemma 14, we get that $\tanh(\gamma) = O(N^{10}d)$ with probability at least $1 - O(N^{-10})$. Therefore, when $t \geq (C/\delta)\log\frac{NdH}{\epsilon}$, we have

$$\left\|\hat{Q}_t\hat{Q}_t^\top - U_\ell U_\ell^\top\right\|_2 \leq \epsilon/H.$$

Therefore, by triangle's inequality,

$$\begin{aligned}
\left\|Q_tQ_t^\top x_i - x_i\right\|_2 &\leq \left\|\hat{Q}_t\hat{Q}_t^\top x_i - x_i\right\|_2 \\
&\leq \left\|U_\ell U_\ell^\top x_i - x_i\right\|_2 + \left\|\left(\hat{Q}_t\hat{Q}_t^\top - U_\ell U_\ell^\top\right) x_i\right\|_2 \\
&\leq \sqrt{\epsilon\left(1 + (1+\delta)^k\right)} + \epsilon.
\end{aligned}$$

Finally, choosing $\delta = 1/k$ and noting that $(1+\delta)^t \leq e$, we get the desired conclusions. $\qquad\square$

Applying Lemma 3 and Lemma 5 and invoking the assumption that $T_1 \gtrsim k\log(Nd)$, we have with probability at least $1 - O(1/N)$,

$$\left\|\left(\hat{U}_t\hat{U}_t^\top - I\right)\sqrt{p_j}\Sigma_j\left(\theta_j^* - \theta_t\right)\right\|_2^2 \leq O\left(\left(\delta_t^2 + \sigma^2\right)\xi_1\right),$$

or equivalently,

$$\left\|\hat{U}_t^\top \Sigma_j\left(\theta_j^* - \theta_t\right)\right\|_2^2 \geq \left\|\Sigma_j\left(\theta_j^* - \theta_t\right)\right\|_2^2 - O\left(\left(\delta_t^2 + \sigma^2\right)\xi_1/p_j\right). \tag{26}$$

### D.5 Proof of Proposition 2

Similar to the proof of Proposition 1, for ease of exposition, we fix an anchor client $i$ and omit the subscript $i$ for simplicity. We further assume client $i$ belongs to cluster $j$, i.e., $z_i = j$. Note that crucially, the global data points on clients $\mathcal{S}_t$ are independent from the local data points on $\mathcal{D}_t$. Thus, in the following analysis, we further condition on $\hat{U}_t$. Then

$$\mathbb{E}[A_t] = \hat{U}_t^\top \Sigma_j\left(\theta_j^* - \theta_t\right)\left(\theta_j^* - \theta_t\right)^\top \Sigma_j\hat{U}_t.$$

**Lemma 6.** *With probability at least $1 - 3N^{-10}$,*

$$\|A_t - \mathbb{E}[A_t]\|_2 \leq O\left(\left(\left\|\theta_j^* - \theta_t\right\|_2^2 + \sigma^2\right)\xi_2\right),$$

*where $\xi_2 = \sqrt{\frac{k}{\ell}\log N} + \frac{k}{\ell}\log^3 N$.*

*Proof.* Note that

$$A_t - \mathbb{E}[A_t] = \frac{1}{\ell}\sum_{j\in\mathcal{D}_t}\hat{U}_t^\top\left(\varepsilon_j\tilde{\varepsilon}_j^\top - \mathbb{E}\left[\varepsilon_j\tilde{\varepsilon}_j^\top\right]\right)\hat{U}_t,$$

where $\varepsilon_j = (y_j - \phi(x_j))\phi(x_j)$ and $\tilde{\varepsilon}_j = (\tilde{y}_j - \phi(\tilde{x}_j))\phi(\tilde{x}_j)$. Let $a_j = \hat{U}_t^\top \varepsilon_j/\sqrt{\|\theta_j^* - \theta_j\|_2^2 + \sigma^2}$ and $b_j = \hat{U}_t^\top \tilde{\varepsilon}_j/\sqrt{\|\theta_j^* - \theta_j\|_2^2 + \sigma^2}$. The rest of the proof follows analogously as that of Lemma 3. $\qquad\square$

Applying Lemma 6 and Lemma 5, when $T_2 \gtrsim \log(Nd)$, we have with probability at least $1 - O(N^{-10})$

$$\left| \hat{\beta}_t^\top \hat{U}_t^\top \Sigma_j \left( \theta_j^* - \theta_t \right) \right|^2 \geq \left\| \hat{U}_t^\top \Sigma_j \left( \theta_j^* - \theta_t \right) \right\|_2^2 - O\left( \left( \left\| \theta_j^* - \theta_t \right\|_2^2 + \sigma^2 \right) \xi_2 \right).$$

Applying Proposition 1, we have with probability at least $1 - O(N^{-10})$

$$\left\| \hat{U}_t^\top \Sigma_j \left( \theta_j^* - \theta_t \right) \right\|_2^2 \geq \left\| \Sigma_j \left( \theta_j^* - \theta_t \right) \right\|_2^2 - O\left( \left( \delta_t^2 + \sigma^2 \right) \xi_1 / p_j \right).$$

Let $\mathcal{E}_t$ denote the event such that the above two displayed equations hold simultaneously. Then $\mathbb{P}\{\mathcal{E}_t\} \geq 1 - O(N^{-10})$. In the following, we assume event $\mathcal{E}_t$ holds.

Combining the last two displayed equations yields that

$$\left\| \Sigma_j \left( \theta_j^* - \theta_t \right) \right\|_2^2 - O\left( \left( \delta_t^2 + \sigma^2 \right) \left( \xi_1 / p_j + \xi_2 \right) \right) \leq \left| \hat{\beta}_t^\top \hat{U}_t^\top \Sigma_j \left( \theta_j^* - \theta_t \right) \right|^2 \leq \left\| \Sigma_j \left( \theta_j^* - \theta_t \right) \right\|_2^2.$$

Moreover, since

$$\hat{\sigma}_t^2 = \hat{\beta}_t^\top \hat{U}_t^\top A_t \hat{U}_t \hat{\beta}_t = \hat{\beta}_t^\top \hat{U}_t^\top \mathbb{E}\left[ A_t \right] \hat{U}_t \hat{\beta}_t + \hat{\beta}_t^\top \hat{U}_t^\top \left( A_t - \mathbb{E}\left[ A_t \right] \right) \hat{U}_t \hat{\beta}_t,$$

it follows that

$$\left| \hat{\sigma}_t^2 - \left| \hat{\beta}_t^\top \hat{U}_t^\top \Sigma_j \left( \theta_j^* - \theta_t \right) \right|^2 \right| \leq O\left( \left( \delta_t^2 + \sigma^2 \right) \xi_2 \right).$$

Combining the last two displayed equations yields that

$$\left| \hat{\sigma}_t^2 - \left\| \Sigma_j \left( \theta_j^* - \theta_t \right) \right\|_2^2 \right| \leq O\left( \left( \delta_t^2 + \sigma^2 \right) \left( \xi_1 / p_j + \xi_2 \right) \right).$$

This proves (19).

Under condition (20), we have

$$\left| \hat{\beta}_t^\top \hat{U}_t^\top \Sigma_j \left( \theta_j^* - \theta_t \right) \right|^2 \geq \left( 1 - \frac{\alpha^2}{64\beta^2} \right) \left\| \Sigma_j \left( \theta_j^* - \theta_t \right) \right\|_2^2 \tag{27}$$

and

$$\left( 1 - \frac{\alpha^2}{32\beta^2} \right) \left\| \Sigma_j \left( \theta_j^* - \theta_t \right) \right\|_2^2 \leq \hat{\sigma}_t^2 \leq \left( 1 + \frac{\alpha^2}{32\beta^2} \right) \left\| \Sigma_j \left( \theta_j^* - \theta_t \right) \right\|_2^2 \tag{28}$$

Now we show that $\theta_t$ converges to $\theta_j^*$. Note that

$$\left( \theta_j^* - \theta_{t+1} \right)^\top \Sigma_j^2 \left( \theta_j^* - \theta_{t+1} \right) = \left( \theta_j^* - \theta_t \right)^\top \Sigma_j^2 \left( \theta_j^* - \theta_t \right) - 2\eta_t \left( \theta_j^* - \theta_t \right)^\top \Sigma_j^2 r_t + \eta_t^2 r_t^\top \Sigma_j^2 r_t.$$

In view of (27), and recalling $r_t = \hat{U}_t \hat{\beta}_t$, we have $\|r_t\|_2 = 1$ and under condition (20)

$$\left\langle r_t, \Sigma_j \left( \theta_j^* - \theta_t \right) \right\rangle^2 \geq \left( 1 - \frac{\alpha^2}{64\beta^2} \right) \left\| \Sigma_j \left( \theta_j^* - \theta_t \right) \right\|_2^2.$$

We decompose

$$\Sigma_j \left( \theta_j^* - \theta_t \right) = a_t r_t + b_t r_t^\perp,$$

for some unit vector $r_t^\perp$ that is perpendicular to $r_t$. Since $a_t^2 + b_t^2 = \|\Sigma_j(\theta_j^* - \theta_t)\|_2^2$, we have $|b_t| \leq \frac{\alpha}{8\beta} \|\Sigma_j(\theta_j^* - \theta_t)\|_2$. Hence,

$$\begin{aligned} \left( \theta_j^* - \theta_t \right)^\top \Sigma_j^2 r_t &= \left( a_t r_t + b_t r_t^\perp \right)^\top \Sigma_j r_t \\ &\geq a_t \alpha - |b_t| \beta \\ &\geq \sqrt{1 - \frac{\alpha^2}{64\beta^2}} \, \alpha \left\| \Sigma_j(\theta_j^* - \theta_t) \right\|_2 - \frac{\alpha}{8} \left\| \Sigma_j(\theta_j^* - \theta_t) \right\|_2 \\ &\geq \frac{\alpha}{2} \left\| \Sigma_j(\theta_j^* - \theta_t) \right\|_2, \end{aligned}$$

where $\lambda_{\min}(\Sigma_j) \geq \alpha$ and $\beta \geq \max_{j \in [k]} \|\Sigma_j\|_2$. It follows that

$$\left(\theta_j^* - \theta_{t+1}\right)^\top \Sigma_j^2 \left(\theta_j^* - \theta_{t+1}\right) \leq \left(\theta_j^* - \theta_t\right)^\top \Sigma_j^2 \left(\theta_j^* - \theta_t\right) - \eta_t \alpha \left\|\Sigma_j \left(\theta_j^* - \theta_t\right)\right\|_2 + \eta_t^2 \|\Sigma_j\|_2^2.$$

Recall the choice of step size $\eta_t = \alpha \hat{\sigma}_t / (2\beta^2)$. In view of (28), we get that

$$\left(\theta_j^* - \theta_{t+1}\right)^\top \Sigma_j^2 \left(\theta_j^* - \theta_{t+1}\right) \leq \left(\theta_j^* - \theta_t\right)^\top \Sigma_j^2 \left(\theta_j^* - \theta_t\right) - \frac{\alpha^2}{4\beta^2} \left\|\Sigma_j \left(\theta_j^* - \theta_t\right)\right\|_2 \hat{\sigma}_t$$

$$\leq \left(1 - \frac{\alpha^2}{8\beta^2}\right) \left\|\Sigma_j \left(\theta_i^* - \theta_t\right)\right\|_2^2,$$

Therefore,

$$\left\|\Sigma_j \left(\theta_j^* - \theta_{t+1}\right)\right\|_2^2 \leq \left(1 - \frac{\alpha^2}{8\beta^2}\right) \left\|\Sigma_j \left(\theta_j^* - \theta_t\right)\right\|_2^2,$$

This proves (21).

# E    Analysis of Phase 2

Throughout the proof in this section, we assume without loss of generality that the optimal permutation in (6) is identity.

## E.1    Derivation of global iteration

*Proof of Lemma 1.* We first prove the result for FedAvg. By definition, we have

$$\nabla_j L_i(\theta) = \frac{\lambda_{ij,t}}{n_i} \phi(\boldsymbol{x}_i)^\top (\phi(\boldsymbol{x}_i)\theta_j - y_i),$$

where $\lambda_{ij,t} = \mathbb{1}\{j = z_{i,t}\}$ and $\nabla_j$ denotes the gradient with respect to $\theta_j$. Then the one-step local gradient descent at client $i$ is

$$[\mathcal{G}_i(\theta)]_j = \begin{cases} \theta_j, & j \neq z_{i,t}, \\ g_i(\theta_j) \triangleq \theta_j - \eta_i \phi(\boldsymbol{x}_i)^\top (\phi(\boldsymbol{x}_i)\theta - y_i), & j = z_{i,t}, \end{cases}$$

where $\eta_i = \eta / n_i$. Iterating $s$ steps yields that [SXY21]

$$g_i^s(\theta_j) = (I - \eta_i \phi(\boldsymbol{x}_i)^\top \phi(\boldsymbol{x}_i))^s \theta_j + \sum_{\ell=0}^{s-1} (I - \eta_i \phi(\boldsymbol{x}_i)^\top \phi(\boldsymbol{x}_i))^\ell \eta_i \phi(\boldsymbol{x}_i)^\top y_i$$

$$\stackrel{(a)}{=} \theta_j - \sum_{\ell=0}^{s-1} (I - \eta_i \phi(\boldsymbol{x}_i)^\top \phi(\boldsymbol{x}_i))^\ell \eta_i \phi(\boldsymbol{x}_i)^\top (\phi(\boldsymbol{x}_i)\theta_j - y_i)$$

$$\stackrel{(b)}{=} \theta_j - \eta_i \phi(\boldsymbol{x}_i)^\top P_i (\phi(\boldsymbol{x}_i)\theta_j - y_i),$$

where $(a)$ used $I - (I - X)^s = \sum_{\ell=0}^{s} (I - X)^\ell X$, and $(b)$ used $(I - X^\top X)^\ell X^\top = X^\top (I - XX^\top)^\ell$ and the definition of $P_i$. Then,

$$\theta_{ij,t} = [\mathcal{G}_i^s(\theta_{t-1})]_j = \lambda_{ij,t} g_i^s(\theta_{j,t-1}) + (1 - \lambda_{ij,t})\theta_{j,t-1}$$

$$= \theta_{j,t-1} - \eta_i \lambda_{ij,t} \phi(\boldsymbol{x}_i)^\top P_i (\phi(\boldsymbol{x}_i)\theta_{j,t-1} - y_i).$$

We obtain the global iteration:

$$\theta_{j,t} = \sum_{i=1}^{M} \frac{n_i}{N} \theta_{ij,t} = \theta_{j,t-1} - \frac{\eta}{N} \sum_{i=1}^{M} \lambda_{ij,t} \phi(\boldsymbol{x}_i)^\top P_i (\phi(\boldsymbol{x}_i)\theta_{j,t-1} - y_i),$$

which is (11) using matrix notations.

The proof for FedProx is similar. The first order condition for the local proximal optimization is

$$\eta_i \lambda_{ij,t} \phi(\boldsymbol{x}_i)^\top (\phi(\boldsymbol{x}_i)\theta_{ij,t} - y_i) + (\theta_{ij,t} - \theta_{j,t-1}) = 0, \quad j \in [k].$$

Therefore, if $j \neq z_{i,t}$, then $\theta_{ij,t} = \theta_{j,t-1}$; if $j = z_{i,t}$, then

$$
\begin{aligned}
\theta_{ij,t} &= (I + \eta_i \phi(\boldsymbol{x}_i)^\top \phi(\boldsymbol{x}_i))^{-1}(\theta_{j,t-1} + \eta_i \phi(\boldsymbol{x}_i)^\top y_i) \\
&\overset{(a)}{=} \theta_{j,t-1} - \eta_i (I + \eta_i \phi(\boldsymbol{x}_i)^\top \phi(\boldsymbol{x}_i))^{-1} \phi(\boldsymbol{x}_i)^\top (\phi(\boldsymbol{x}_i)\theta_{j,t-1} - y_i) \\
&\overset{(b)}{=} \theta_{j,t-1} - \eta_i \phi(\boldsymbol{x}_i)^\top P_i(\phi(\boldsymbol{x}_i)\theta_{j,t-1} - y_i),
\end{aligned}
$$

where $(a)$ used $I - (I+X)^{-1} = (I+X)^{-1}X$, and $(b)$ used $(I+X^\top X)^{-1}X^\top = X^\top (I+XX^\top)^{-1}$ and the definition of $P_i$. The remaining steps are the same as those in FedAvg. $\square$

## E.2 Convergence analysis of Phase 2

We analyze the three terms on the right-hand side of (12) separately. The first term of (12) is the main term due to the decreasing of estimation error, and the last term is the stochastic variation due to the observation noise $\zeta$. We have the following lemmas on the eigenvalues of $K_j$ and the concentration of the observation noise.

**Lemma 7.** *There exists constants $c$ and $C$ such that, with probability $1 - 2k e^{-d}$,*

$$
c\alpha \frac{sN_j}{\kappa N} \leq \lambda_{\min}(K_j) \leq \lambda_{\max}(K_j) \leq C\beta \frac{sN_j}{N}, \quad \forall j \in [k].
$$

*Proof.* Since $\phi(\boldsymbol{x}_{I_j})$ of size $N_j \times d$ consists of independent and sub-Gaussian rows, by a covering argument [Ver18, Theorem 4.6.1], with probability $1 - 2e^{-d}$,

$$
\alpha N_j - C(\sqrt{dN_j} \vee d) \leq \sigma_{\min}^2(\phi(\boldsymbol{x}_{I_j})) \leq \sigma_{\max}^2(\phi(\boldsymbol{x}_{I_j})) \leq \beta N_j + C(\sqrt{dN_j} \vee d),
$$

where $\sigma_{\max}$ and $\sigma_{\min}$ denote the largest and smallest singular values, respectively, and $C$ is an absolute constant. By definition, $K_j = \frac{1}{N}\phi(\boldsymbol{x}_{I_j})^\top P_{I_j}\phi(\boldsymbol{x}_{I_j})$, where $P_{I_j}$ is a symmetric matrix. It is shown in [SXY21, Lemma 3] that

$$
s/\kappa \leq \lambda_{\min}(P_{I_j}) \leq \lambda_{\max}(P_{I_j}) \leq s.
$$

The conclusion follows from the condition $N_j \gtrsim d$ and a union bound over $j \in [k]$. $\square$

**Lemma 8.** *Given the input features $\phi(\boldsymbol{x})$, there exists a constant $C$ such that with probability at least $1 - k\exp(-d)$,*

$$
\|B\Lambda_j\zeta\|_2^2 \leq C \frac{\sigma^2 sd}{N}\|K_j\|_2, \quad \forall j \in [k].
$$

*Proof.* Note that
$$
\|B\Lambda_j\zeta\|_2^2 = \zeta^\top \Lambda_j B^\top B\Lambda_j\zeta = \langle \Lambda_j B^\top B\Lambda_j, \zeta\zeta^\top \rangle.
$$
Since $\mathbb{E}\left[\zeta\zeta^\top\right] \preceq \sigma^2 I$, it follows that

$$
\mathbb{E}\left[\|B\Lambda_j\zeta\|_2^2\right] = \mathbb{E}\left[\langle \Lambda_j B^\top B\Lambda_j, \zeta\zeta^\top \rangle\right] \leq \sigma^2 \mathsf{Tr}\left(\Lambda_j B^\top B\Lambda_j\right) = \sigma^2 \mathsf{Tr}\left(B\Lambda_j^2 B^\top\right).
$$

Recall that

$$
B\Lambda_j^2 B^\top = \frac{1}{N^2}\phi(\boldsymbol{x}_{I_j})^\top P_{I_j}^2 \phi(\boldsymbol{x}_{I_j}) \overset{(a)}{\preceq} \frac{s}{N^2}\phi(\boldsymbol{x}_{I_j})^\top P_{I_j}\phi(\boldsymbol{x}_{I_j}) = \frac{s}{N}K_j, \tag{29}
$$

where $(a)$ holds because $\|P_{I_j}\|_2 \leq s$. Therefore,

$$
\mathbb{E}\left[\|B\Lambda_j\zeta\|_2^2\right] = \mathbb{E}\left[\langle \Lambda_j B^\top B\Lambda_j, \zeta\zeta^\top \rangle\right] \leq \frac{\sigma^2 sd}{N}\|K_j\|_2.
$$

Next, using Hanson-Wright's inequality [RV+13], we get

$$
\mathbb{P}\left\{\langle \Lambda_j B^\top B\Lambda_j, \zeta\zeta^\top \rangle - \mathbb{E}\left[\langle \Lambda_j B^\top B\Lambda_j, \zeta\zeta^\top \rangle\right] \geq \delta\right\}
$$
$$
\leq \exp\left(-c_1 \min\left\{\frac{\delta}{\sigma^2\|\Lambda_j B^\top B\Lambda_j\|_2}, \frac{\delta^2}{\sigma^4\|\Lambda_j B^\top B\Lambda_j\|_F^2}\right\}\right),
$$

where $c_1 > 0$ is a universal constant. Note that

$$\|\Lambda_j B^\top B \Lambda_j\|_2 = \|B \Lambda_j^2 B^\top\|_2 \leq \frac{s}{N}\|K_j\|_2,$$

$$\|\Lambda_j B^\top B \Lambda_j\|_{\mathrm{F}} = \|B \Lambda_j^2 B^\top\|_{\mathrm{F}} \leq s\|K_j\|_{\mathrm{F}} \leq \frac{s\sqrt{d}}{N}\|K_j\|_2.$$

Therefore, by choosing $\delta = C\frac{\sigma^2 sd}{N}\|K_j\|_2$ for a sufficiently large constant $C$, we get that with probability at least $1 - \exp(-d)$,

$$\langle \Lambda_j B^\top B \Lambda_j, \zeta\zeta^\top \rangle \leq \mathbb{E}\left[\langle \Lambda_j B^\top B \Lambda_j, \zeta\zeta^\top \rangle\right] + \delta \leq (C+1)\,\sigma^2 \frac{sd}{N}\|K_j\|_2.$$

The conclusion follows from a union bound over all $j \in [k]$. $\qquad\square$

Combining Lemmas 2, 7, and 8, next we prove Theorem 2.

*Proof of Theorem 2.* We prove the result conditioning on the high probability events in Lemmas 2, 7, and 8 that happen with probability at least $1 - Cke^{-d}$. We obtain from Lemma 7 that

$$\|I - \eta K_j\|_2 \leq 1 - C\eta s\rho/\kappa.$$

Combining Lemmas 7 and 8 yields

$$\|B\Lambda_j \zeta\|_2 \lesssim s\sigma\sqrt{\frac{d}{N}}.$$

Plugging the above upper bounds and Lemma 2 into (12), we get

$$\|\theta_{j,t} - \theta_j^*\|_2 \leq \left(1 - C\eta s\left(\frac{\rho}{\kappa} - \nu\log\frac{e}{\nu}\right)\right)d(\theta_{t-1}, \theta^*) + C\eta s\sigma\left(\sqrt{\frac{d}{N}} + \nu\log\frac{e}{\nu}\right), \quad \forall j \in [k].$$

Since $\nu\log(e/\nu) \lesssim \rho/\kappa$ and $\nu \gtrsim \sqrt{d/N}$, we conclude (9).

Let $\hat{\theta}_{i,t} = \theta_{z_{i,t},t}$ be client $i$'s estimate of its own model parameter. If client $i$ is clustered correctly such that $z_{i,t} = z_i$, where the success probability $\mathbb{P}\{z_{i,t} = z_i\}$ is shown in Lemma 10 (which can be found in Appendix 2), it follows from (9) that, for $t \geq T + 1$,

$$\|\hat{\theta}_{i,t} - \theta_{z_i}^*\|_2 \leq d(\theta_t, \theta^*) \leq (1 - C_1 s\eta\rho/\kappa)^{t-T} d(\theta_T, \theta^*) + \frac{C_2}{C_1}\frac{\sigma\kappa}{\rho}\nu\log\frac{e}{\nu}.$$

The proof is completed. $\qquad\square$

### E.2.1 Proof of Lemma 2

This subsection is devoted to the proof of Lemma 2 using the following road map:

$$d(\theta_t, \theta^*) \downarrow \implies \sum_{i:i\in S_{j,t}} n_i \downarrow \implies \|\phi(\boldsymbol{x}_{S_{j,t}})\|_2, \|\zeta_{S_{j,t}}\|_2 \downarrow \implies \|B\mathcal{E}_{j,t}(\phi(\boldsymbol{x})\theta_{j,t-1} - y)\|_2 \downarrow.$$

Specifically, a small estimation error $d(\theta_t, \theta^*)$ implies an upper bound on the total number of incorrectly clustered data points $\sum_{i \in S_{j,t}} n_i$; then we upper bound $\|\phi(\boldsymbol{x}_{S_j^t})\|_2$ and $\|\zeta_{S_j^t}\|_2$ using sub-Gaussian concentration and the union bound; finally we conclude the result from (15).

We first upper bound $\sum_{i \in S_{j,t}} n_i$. Using (8), the set $S_{j,t} = I_j \ominus I_{j,t}$ is equivalently the union of

$$I_j - I_{j,t} = \left\{i \in I_j : \|y_i - \phi(\boldsymbol{x}_i)\theta_{j,t-1}\|_2 \geq \min_{\ell \neq j}\|y_i - \phi(\boldsymbol{x}_i)\theta_{\ell,t-1}\|_2\right\},$$

$$I_{j,t} - I_j = \left\{i \notin I_j : \|y_i - \phi(\boldsymbol{x}_i)\theta_{j,t-1}\|_2 \leq \min_{\ell \neq j}\|y_i - \phi(\boldsymbol{x}_i)\theta_{\ell,t-1}\|_2\right\}.$$

Therefore, $S_{j,t} = S_j(\theta_{t-1})$, where $S_j$ is defined in (16). The next lemma upper bounds the VC dimensions of the binary function classes specified in (17).

**Lemma 9.** *For $k \geq 2$, the VC dimensions of $\mathcal{F}_j^{\mathrm{I}}$ and $\mathcal{F}_j^{\mathrm{II}}$ are at most $O(dk \log k)$.*

*Proof.* We focus on the proof for $\mathcal{F}_j^{\mathrm{I}}$ for a fixed $j \in [k]$, and the proof for $\mathcal{F}_j^{\mathrm{II}}$ is similar. We count the number of faces in the arrangement of geometric objects, which is also known as the number of sign patterns. Specifically, here we define the sign patterns of binary functions $g_1(\theta), \ldots, g_m(\theta)$ as the set

$$\left\{ (g_1(\theta), \ldots, g_m(\theta)) : \theta \in \mathbb{R}^{dk} \right\}.$$

Suppose $\mathcal{F}_j^{\mathrm{I}}$ shatters $m$ points denoted by $(\boldsymbol{x}_1, y_1), \ldots, (\boldsymbol{x}_m, y_m)$. Define binary functions

$$q_{i,\ell}(\theta) \triangleq \mathbb{1}\{P_{\ell,j}[\boldsymbol{x}_i, y_i](\theta) \geq 0\}, \quad g_i(\theta) \triangleq \max_{\ell \neq j} q_{i,\ell}(\theta).$$

It is necessary that the number of sign patterns of $g_1(\theta), \ldots, g_m(\theta)$ is $2^m$. Note that every $P_{\ell,j}[\boldsymbol{x}_i, y_i]$ is a $(dk)$-variate quadratic function. By the Milnor-Thom theorem (see, e.g., [Mat13, Theorem 6.2.1]), if $m(k-1) \geq dk \geq 2$, the number of sign patterns of $m(k-1)$ binary functions $q_{1,\ell}, \ldots, q_{m,\ell}$ for $\ell \neq j$ is at most $(\frac{100m(k-1)}{dk})^{dk}$. Since each $g_i$ is the maximum of $q_{i,\ell}$ over $\ell \neq j$, the number of sign patterns of $g_1, \ldots, g_m$ is upper bounded by $(\frac{100m(k-1)}{dk})^{dk}$. Consequently, we obtain $2^m \leq (\frac{100m(k-1)}{dk})^{dk}$, and hence $m \lesssim dk \log k$. If instead, $m(k-1) < dk$, then the conclusion $m \lesssim dk \log k$ trivially holds. $\qquad\square$

Next we show the uniform deviation of the incorrectly clustered data points. Due to the quantity skew, we consider a weighted empirical process $G_j(\theta) = \sum_{i=1}^{M} n_i \mathbb{1}\{i \in S_j(\theta)\}$. Since the local data $(\boldsymbol{x}_i, y_i)$ on different clients are independent, for a fixed $\theta$, the events $\{i \in S_j(\theta)\}$ as functions of $(\boldsymbol{x}_i, y_i)$ are mutually independent. Using the binary function classes in (17), we have

$$\mathbb{E}\left[\sup_\theta |G_j(\theta) - \mathbb{E}[G_j(\theta)]|\right]$$

$$\leq \mathbb{E}\left[\sup_{f \in \mathcal{F}_j^{\mathrm{I}}} \left|\sum_{i \in I_j} n_i(f(\boldsymbol{x}_i, y_i) - \mathbb{E}[f(\boldsymbol{x}_i, y_i)])\right|\right] + \mathbb{E}\left[\sup_{f \in \mathcal{F}_j^{\mathrm{II}}} \left|\sum_{i \notin I_j} n_i(f(\boldsymbol{x}_i, y_i) - \mathbb{E}[f(\boldsymbol{x}_i, y_i)])\right|\right]$$

$$\lesssim \sqrt{dk \log k \sum_{i \in \mathcal{I}_j} n_i^2} + \sqrt{dk \log k \sum_{i \notin \mathcal{I}_j} n_i^2}$$

$$\leq \sqrt{2dk \log k \sum_{i=1}^{M} n_i^2}, \tag{30}$$

where the second inequality follows from the uniform deviation of weighted empirical processes in Lemma 12 and the upper bound of VC dimensions in Lemma 9. Finally, we use the McDiarmid's inequality to establish a high-probability tail bound. Note that we can write

$$\sup_\theta |G_j(\theta) - \mathbb{E}[G_j(\theta)]| \triangleq h(Z_1, \ldots, Z_M)$$

as a function $h$ of $Z_i = (\boldsymbol{x}_i, y_i)$ with bounded differences: for any $i, z_i, z_i'$,

$$|h(z_1, \ldots, z_i, \ldots, z_M) - h(z_1, \ldots, z_i', \ldots, z_M)| \leq n_i.$$

By McDiarmid's inequality, we have

$$\mathbb{P}\{h(Z_1, \ldots, Z_M) - \mathbb{E}[h(Z_1, \ldots, Z_M)] \geq t\} \leq \exp\left(-\frac{2t^2}{\sum_{i=1}^{M} n_i^2}\right). \tag{31}$$

Therefore, combining (30) and (31), and by a union bound, with probablity at least $1 - k^{-dk}$,

$$\sup_\theta |G_j(\theta) - \mathbb{E}[G_j(\theta)]| \lesssim \sqrt{dk \log k \sum_{i=1}^{M} n_i^2} = N\sqrt{\frac{dk \log k}{M}}(\chi^2(n) + 1), \quad \forall j \in [k]. \tag{32}$$

**Lemma 10.** *Suppose $\epsilon \leq \frac{\sqrt{\alpha/\beta}}{3}\Delta$. Then,*

$$\sup_{\theta:d(\theta,\theta^*)\leq\epsilon} \mathbb{P}[i \in S_j(\theta)] \leq 4k \exp\left(-cn_i\alpha^2\left(1 \wedge \frac{\Delta^2}{\sigma^2}\right)^2\right), \quad \forall j \in [k],$$

*where $c$ is an absolute constant.*

*Proof.* For $i \in I_j$, it follows from (16) and the union bound that

$$\mathbb{P}\{i \in S_j(\theta)\} \leq \sum_{\ell \neq j} \mathbb{P}\{\|y_i - \phi(\boldsymbol{x}_i)\theta_j\|_2 \geq \|y_i - \phi(\boldsymbol{x}_i)\theta_\ell\|_2\}$$

$$= \sum_{\ell \neq j} \mathbb{P}\{\|\phi(\boldsymbol{x}_i)(\theta_j^* - \theta_j) + \zeta_i\|_2 \geq \|\phi(\boldsymbol{x}_i)(\theta_j^* - \theta_\ell) + \zeta_i\|_2\}. \tag{33}$$

For any $u \in \mathbb{R}^d$, the $n_i$-dimensional random vector $\phi(\boldsymbol{x}_i)u + \zeta_i$ has independent and $(\|u\|_2^2 + \sigma^2)$-sub-Gaussian coordinates. Applying Bernstein inequality yields that

$$\mathbb{P}\left\{\left|\frac{1}{n_i}\|\phi(\boldsymbol{x}_i)u + \zeta_i\|_2^2 - \left(\mathbb{E}[\zeta_{i1}^2] + \|u\|_{\Sigma_i}^2\right)\right| \geq (\|u\|_2^2 + \sigma^2)t\right\} \leq 2\exp\left(-cn_i(t \wedge t^2)\right), \tag{34}$$

where $\Sigma_i = \mathbb{E}[\phi(x_{i1})\phi(x_{i1})^\top]$. Let $u_1 \triangleq \theta_j^* - \theta_j$ and $u_2 \triangleq \theta_j^* - \theta_\ell$. By assumptions that $\|\theta_\ell - \theta_\ell^*\|_2 \leq \epsilon$ for all $\ell \in [k]$ and $\|\theta_j^* - \theta_\ell^*\|_2 \geq \Delta$ for $\ell \neq j$, we have $\|u_1\|_2 \leq \epsilon$, $\|u_2\|_2 \geq \Delta - \epsilon$. Applying the condition $\epsilon \leq \frac{1}{3\sqrt{\beta/\alpha}}\Delta \leq \frac{1}{3}\Delta$, we get

$$\|u_2\|_{\Sigma_i}^2 - \|u_1\|_{\Sigma_i}^2 \geq \alpha(\Delta - \epsilon)^2 - \beta\epsilon^2 \geq \alpha\Delta^2/3. \tag{35}$$

Therefore, let $m = \mathbb{E}[\zeta_{i1}^2] + (1-p)\|u_1\|_{\Sigma_i}^2 + p\|u_2\|_{\Sigma_i}^2$ with $p = \frac{\|u_1\|_2^2 + \sigma^2}{\|u_1\|_2^2 + \|u_2\|_2^2 + 2\sigma^2}$, and we obtain from (34) that

$$\mathbb{P}\{\|\phi(\boldsymbol{x}_i)(\theta_j^* - \theta_j) + \zeta_i\|_2 \geq \|\phi(\boldsymbol{x}_i)(\theta_j^* - \theta_\ell) + \zeta_i\|_2\}$$

$$\leq \mathbb{P}\left\{\frac{1}{n_i}\|\phi(\boldsymbol{x}_i)u_1 + \zeta_i\|_2^2 \geq m\right\} + \mathbb{P}\left\{\frac{1}{n_i}\|\phi(\boldsymbol{x}_i)u_2 + \zeta_i\|_2^2 \leq m\right\}$$

$$\leq 4\exp\left(-cn_i(t \wedge t^2)\right),$$

where $t = \frac{\|u_2\|_{\Sigma_i}^2 - \|u_1\|_{\Sigma_i}^2}{\|u_1\|_2^2 + \|u_2\|_2^2 + 2\sigma^2} \gtrsim \alpha(1 \wedge \frac{\Delta^2}{\sigma^2})$ using the lower bound of seperation in (35). We conclude the proof for $i \in I_j$ from (33). Similarly, for $i \in I_\ell$ with $\ell \neq j$, we have

$$\mathbb{P}\{i \in S_j(\theta)\} \leq \mathbb{P}\{\|\phi(\boldsymbol{x}_i)(\theta_\ell^* - \theta_\ell) + \zeta_i\|_2 \geq \|\phi(\boldsymbol{x}_i)(\theta_\ell^* - \theta_j) + \zeta_i\|_2\}.$$

The conclusion follows from a similar argument. $\qquad\square$

Let $N_I \triangleq \sum_{i \in I} n_i$ denote the total number of data in a subset of clients $I \subseteq [M]$. It follows from (32) and Lemma 10 that, with probability $1 - k^{-dk}$,

$$N_{S_{j,t}} = \sum_{i \in S_{j,t}} n_i \leq \nu N, \tag{36}$$

where $\nu$ is defined in (10). Conditioning on total number of incorrectly clustered data points $N_I$, the next lemma upper bounds $\|\phi(\boldsymbol{x}_I)\|_2$ and $\|\zeta_I\|_2$.

**Lemma 11.** *With probability $1 - 4e^{-d}$,*

$$\sup_{N_I \leq \nu N} \frac{1}{N}\|\phi(\boldsymbol{x}_I)\|_2^2 \lesssim \nu \log\frac{e}{\nu}, \qquad \sup_{N_I \leq \nu N} \frac{1}{N}\|\zeta_I\|_2^2 \lesssim \sigma^2\nu\log\frac{e}{\nu}. \tag{37}$$

*Proof.* Since $\phi(x_{ij})$ are independent and sub-Gaussian random vectors in $\mathbb{R}^d$, for a fixed $I \subseteq [M]$, with probability at least $1 - 2e^{-t}$,

$$\|\phi(\boldsymbol{x}_I)\|_2^2 \leq \beta N_I + C'\left(\sqrt{(d+t)N_I} + (d+t)\right),$$

for some absolute constant $C' > 0$. There are at most $\binom{N}{\nu N} \leq \exp(N\nu \log(e/\nu))$ many different $I$ with $N_I \leq N'$. Hence, applying the union bound yields that, with probability at least $1 - 2e^{-d}$,

$$\sup_{N_I \leq \nu N} \|\phi(\boldsymbol{x}_I)\|_2^2 \lesssim N\nu \log \frac{e}{\nu},$$

where we used $\nu \gtrsim \frac{d}{N}$. Since $\zeta_{ij}$ are independent and sub-Gaussian with $\mathbb{E}[\zeta_{ij}^2] \leq \sigma^2$, the second inequality in (37) follows from a similar argument. $\square$

Conditioning on the high probability events of (36) and (37), we obtain

$$\|\phi(\boldsymbol{x}_{S_{j,t}})\|_2 \lesssim \sqrt{N\nu \log \frac{e}{\nu}}, \qquad \|\zeta_{S_{j,t}}\|_2 \lesssim \sigma \sqrt{N\nu \log \frac{e}{\nu}}.$$

Since $\|P_{S_{j,t}}\|_2 \leq s$, we conclude from (14) and (15) that

$$
\begin{aligned}
\|B\mathcal{E}_{j,t}(\phi(\boldsymbol{x})\theta_{j,t-1} - y)\|_2 &\leq \frac{1}{N}\|\phi(\boldsymbol{x}_{S_{j,t}})\|_2\|P_{S_{j,t}}\|_2\|\phi(\boldsymbol{x}_{S_{j,t}})\theta_{j,t-1} - y_{S_{j,t}}\|_2 \\
&\lesssim \frac{s}{N}\left(d(\theta_{t-1}, \theta^*)\|\phi(\boldsymbol{x}_{S_{j,t}})\|_2^2 + \|\phi(\boldsymbol{x}_{S_{j,t}})\|_2\|\zeta_{S_{j,t}}\|_2\right) \\
&\lesssim s(d(\theta_{t-1}, \theta^*) + \sigma)\nu \log \frac{e}{\nu}.
\end{aligned}
$$

### E.2.2 Auxiliary lemma

**Lemma 12.** *Consider a weighted empirical process $G_n(f) = \sum_{i=1}^n \lambda_i f(X_i)$ for binary functions $f \in \mathcal{F}$, where $X_i$'s are independent and the VC dimension of $\mathcal{F}$ is at most $d$. Then*

$$\mathbb{E}\left[\sup_{f \in \mathcal{F}} |G_n(f) - \mathbb{E}G_n(f)|\right] \lesssim \sqrt{d \sum_{i=1}^n \lambda_i^2}.$$

*Proof.* Since $X_i$'s are independent, by symmetrization,

$$\mathbb{E}\left[\sup_{f \in \mathcal{F}} |G_n(f) - \mathbb{E}G_n(f)|\right] \leq 2\mathbb{E}\left[\sup_{f \in \mathcal{F}} \left|\sum_{i=1}^n \epsilon_i \lambda_i f(X_i)\right|\right],$$

where $\epsilon_i$ are i.i.d. Rademacher random variables. Next, by conditioning on $X_i$'s, we aim to apply Dudley's integral. Since $\epsilon_i$ are independent and 1-sub-Gaussian, for any $f, g \in \mathcal{F}$, the increment $\sum_i \epsilon_i \lambda_i f(X_i) - \sum_i \epsilon_i \lambda_i g(X_i)$ is also sub-Gaussian with a variance parameter

$$\sum_{i=1}^n \lambda_i^2 (f-g)(X_i)^2 = \left(\sum_{i=1}^n \lambda_i^2\right) \|f - g\|_{L^2(\mu_n)}^2,$$

where $\mu_n$ denotes the weighted empirical measure $\frac{1}{\sum_i \lambda_i^2} \sum_i \lambda_i^2 \delta_{X_i}$. Apply Dudley's integral (see, e.g., [Ver18, Theorem 8.1.3]) conditioning on $X_i$'s, we get that

$$\mathbb{E}\left[\sup_{f \in \mathcal{F}} \left|\sum_{i=1}^n \epsilon_i \lambda_i f(X_i)\right|\right] \lesssim \sqrt{\sum_{i=1}^n \lambda_i^2} \times \mathbb{E}\left[\int_0^1 \sqrt{\log \mathcal{N}(\mathcal{F}, L^2(\mu_n), \epsilon)} d\epsilon\right],$$

where $\mathcal{N}(\mathcal{F}, L^2(\mu_n), \epsilon)$ denotes the $\epsilon$-covering number of $\mathcal{F}$ under $L^2(\mu_n)$. Finally, we can bound the covering number by the VC dimension of $\mathcal{F}$ as (see, e.g., [Ver18, Theorem 8.3.18])

$$\log \mathcal{N}(\mathcal{F}, L^2(\mu_n), \epsilon) \lesssim d \log \frac{2}{\epsilon}.$$

The conclusion follows. $\square$

# F  Truncated matrix Bernstein inequality

**Lemma 13.** *Let $\{a_i : i \in [N]\}$ and $\{b_i : i \in [N]\}$ denote two independent sequences of independent random vectors in $\mathbb{R}^d$. Suppose that $\mathbb{E}\left[\|a_i\|_2^2\right] = O(d)$, $\mathbb{E}\left[\|b_i\|_2^2\right] = O(d)$, $\left\|\mathbb{E}\left[a_i a_i^\top\right]\right\|_2 = O(1)$, $\left\|\mathbb{E}\left[b_i b_i^\top\right]\right\|_2 = O(1)$, and*

$$\mathbb{P}\left\{\|a_i\|_2 \geq t\right\}, \mathbb{P}\left\{\|b_i\|_2 \geq t\right\} \leq \exp\left(-\Omega\left(t/\sqrt{d}\right)\right), \quad \forall t \geq \sqrt{d}.$$

*Let*

$$Y = \sum_{i=1}^N \left(a_i b_i^\top - \mathbb{E}\left[a_i b_i^\top\right]\right).$$

*Then there exists a univeral constant $C > 0$ such that with probability at least $1 - 3\delta$,*

$$\|Y\|_2 \leq C\left(\sqrt{Nd \log \frac{1}{\delta}} + d \log^3(N/\delta)\right).$$

*Proof.* Given $\tau$ to be specified later, define event $\mathcal{E}_i = \{\left\|a_i b_i^\top\right\|_2 \leq \tau\}$. It follows that

$$Y = \sum_{i=1}^N \left(a_i b_i^\top \mathbb{1}\{\mathcal{E}_i\} - \mathbb{E}\left[a_i b_i^\top \mathbb{1}\{\mathcal{E}_i\}\right]\right) + \sum_{i=1}^N a_i b_i^\top \mathbb{1}\{\mathcal{E}_i^c\} - \sum_{i=1}^N \mathbb{E}\left[a_i b_i^\top \mathbb{1}\{\mathcal{E}_i^c\}\right]$$

and hence

$$\|Y\|_2 \leq \left\|\sum_{i=1}^N \left(a_i b_i^\top \mathbb{1}\{\mathcal{E}_i\} - \mathbb{E}\left[a_i b_i^\top \mathbb{1}\{\mathcal{E}_i\}\right]\right)\right\|_2 + \left\|\sum_{i=1}^N a_i b_i^\top \mathbb{1}\{\mathcal{E}_i^c\}\right\|_2 + \left\|\sum_{i=1}^N \mathbb{E}\left[a_i b_i^\top \mathbb{1}\{\mathcal{E}_i^c\}\right]\right\|_2. \tag{38}$$

In the sequel, we bound each term in the RHS separately.

To bound the first term, we will use matrix Bernstein inequality. Let $Y_i = a_i b_i^\top \mathbb{1}\{\mathcal{E}_i\} - \mathbb{E}\left[a_i b_i^\top \mathbb{1}\{\mathcal{E}_i\}\right]$. Then $\mathbb{E}\left[Y_i\right] = 0$ and

$$\|Y_i\|_2 \leq \left\|a_i b_i^\top \mathbb{1}\{\mathcal{E}_i\}\right\|_2 + \left\|\mathbb{E}\left[a_i b_i^\top \mathbb{1}\{\mathcal{E}_i\}\right]\right\|_2 \leq 2\tau.$$

Moreover,

$$\sum_{i=1}^N \mathbb{E}\left[Y_i Y_i^\top\right] = \sum_{i=1}^N \mathbb{E}\left[\left(a_i b_i^\top \mathbb{1}\{\mathcal{E}_i\} - \mathbb{E}\left[a_i b_i^\top \mathbb{1}\{\mathcal{E}_i\}\right]\right)\left(a_i b_i^\top \mathbb{1}\{\mathcal{E}_i\} - \mathbb{E}\left[a_i b_i^\top \mathbb{1}\{\mathcal{E}_i\}\right]\right)^\top\right]$$

$$= \sum_{i=1}^N \left(\mathbb{E}\left[a_i a_i^\top \|b_i\|_2^2 \mathbb{1}\{\mathcal{E}_i\}\right] - \mathbb{E}\left[a_i b_i^\top \mathbb{1}\{\mathcal{E}_i\}\right]\mathbb{E}\left[a_i b_i^\top \mathbb{1}\{\mathcal{E}_i\}\right]^\top\right)$$

Therefore,

$$\sum_{i=1}^N \mathbb{E}\left[Y_i Y_i^\top\right] \preceq \sum_{i=1}^N \mathbb{E}\left[a_i a_i^\top \|b_i\|_2^2 \mathbb{1}\{\mathcal{E}_i\}\right]$$

$$\preceq \sum_{i=1}^N \mathbb{E}\left[a_i a_i^\top \|b_i\|_2^2\right]$$

$$= \sum_{i=1}^N \mathbb{E}\left[\|b_i\|_2^2\right]\mathbb{E}\left[a_i a_i^\top\right] \preceq O(Nd)\mathbf{I}.$$

Moreover, $Y_i Y_i^\top \succeq 0$. Hence, $\left\|\sum_{i=1}^N \mathbb{E}\left[Y_i Y_i^\top\right]\right\|_2 = O(Nd)$. Similarly, we can show that $\left\|\sum_{i=1}^N \mathbb{E}\left[Y_i^\top Y_i\right]\right\|_2 = O(Nd)$. Applying matrix Bernstein inequality [Tro15], we get that with probability at least $1 - \delta$,

$$\left\|\sum_{i=1}^N Y_i\right\|_2 \lesssim \sqrt{Nd \log \frac{1}{\delta}} + \tau \log \frac{1}{\delta}. \tag{39}$$

Next, we bound the second term in (38). Note that on the event $\cap_{i=1}^{N} \mathcal{E}_i$, $\left\| \sum_{i=1}^{N} a_i b_i^{\top} \mathbb{1}\{\mathcal{E}_i^c\} \right\|_2 = 0$.
Note that

$$\mathbb{P}\{\mathcal{E}_i^c\} = \mathbb{P}\left\{ \left\| a_i b_i^{\top} \right\|_2 > \tau \right\} \leq \mathbb{P}\{\|a_i\|_2 \geq \sqrt{\tau}\} + \mathbb{P}\{\|b_i\|_2 \geq \sqrt{\tau}\} \leq 2e^{-\Omega(\sqrt{\tau/d})}.$$

Hence by choosing $\tau = Cd \log^2 \frac{N}{\delta}$ for some sufficiently large constant $C$, we get that $\mathbb{P}\{\mathcal{E}_i^c\} \leq \delta/N$. Thus by union bound,

$$\mathbb{P}\left\{\cap_{i=1}^{N} \mathcal{E}_i\right\} \geq 1 - \sum_{i=1}^{N} \mathbb{P}\{\mathcal{E}_i^c\} \geq 1 - 2\delta. \tag{40}$$

Finally, we bond the third term in (38). Note that

$$\left\| \sum_{i=1}^{N} \mathbb{E}\left[a_i b_i^{\top} \mathbb{1}\{\mathcal{E}_i^c\}\right] \right\|_2 \leq \sum_{i=1}^{N} \left\| \mathbb{E}\left[a_i b_i^{\top} \mathbb{1}\{\mathcal{E}_i^c\}\right] \right\|_2 \leq \sum_{i=1}^{N} \mathbb{E}\left[\left\| a_i b_i^{\top} \mathbb{1}\{\mathcal{E}_i^c\} \right\|_2\right].$$

Moreover,

$$
\begin{aligned}
\mathbb{E}\left[\left\| a_i b_i^{\top} \mathbb{1}\{\mathcal{E}_i^c\} \right\|_2\right] &= \int_0^{\infty} \mathbb{P}\left\{ \left\| a_i b_i^{\top} \mathbb{1}\{\mathcal{E}_i^c\} \right\|_2 \geq t \right\} dt \\
&= \int_0^{\tau} \mathbb{P}\left\{ \left\| a_i b_i^{\top} \right\|_2 \geq \tau \right\} dt + \int_{\tau}^{\infty} \mathbb{P}\left\{ \left\| a_i b_i^{\top} \right\|_2 \geq t \right\} dt \\
&= \tau \frac{\delta}{N} + \int_{\tau}^{\infty} \mathbb{P}\left\{ \left\| a_i b_i^{\top} \right\|_2 \geq t \right\} dt
\end{aligned}
$$

By assumption, for $t \geq \tau = Cd \log^2 \frac{N}{\delta}$,

$$\mathbb{P}\left\{ \left\| a_i b_i^{\top} \right\|_2 \geq t \right\} \leq \mathbb{P}\left\{\|a_i\|_2 \geq \sqrt{t}\right\} + \mathbb{P}\left\{\|b_i\|_2 \geq \sqrt{t}\right\} \leq 2e^{-C'\sqrt{t/d}}$$

for some universal constant $C' > 0$. It follows that

$$\int_{\tau}^{\infty} \mathbb{P}\left\{ \left\| a_i b_i^{\top} \right\|_2 \geq t \right\} dt \leq 2 \int_{\tau}^{\infty} e^{-C'\sqrt{t/d}} dt = 4d\left(\sqrt{\tau/d} + 1/C'\right) e^{-C'\sqrt{\tau/d}},$$

where the equality holds by the identity that $\int_{\tau}^{\infty} e^{-\alpha\sqrt{t}} dt = \frac{2}{\alpha^2}(\sqrt{\tau}\alpha + 1)e^{-\alpha\sqrt{\tau}}$. Therefore,

$$\mathbb{E}\left[\left\| a_i b_i^{\top} \mathbb{1}\{\mathcal{E}_i^c\} \right\|_2\right] \leq \tau \frac{\delta}{N} + 4d\left(\sqrt{\tau/d} + 1/C'\right) e^{-C'\sqrt{\tau/d}} = O\left(\frac{d}{N} \log^2(N/\delta)\right). \tag{41}$$

Plugging (39), (40), and (41) into (38) yields the desired conclusion. $\qquad\square$

## G   Bound on the largest principal angle between random subspaces

Let $U \in \mathbb{R}^{d \times \ell}$ denote an orthogonal matrix and $Q \in \mathbb{R}^{d \times \ell}$ denote a random orthogonal matrix chosen uniformly at random, where $\ell \leq d$.

**Lemma 14.** *With probability at least $1 - O(\epsilon)$,*

$$\sigma_{\min}(U^{\top}Q) \gtrsim \frac{\epsilon}{\sqrt{\ell}(\sqrt{d} + \log(1/\epsilon))}.$$

*Proof.* Since $Q \in \mathbb{R}^{d \times \ell}$ is a random orthogonal matrix, to prove the claim, without loss of generality, we can assume $U = [e_1, e_2, \ldots, e_{\ell}]$, where $e_i$'s are the standard basis vectors in $\mathbb{R}^d$. Let $A \in \mathbb{R}^{d \times \ell}$ denote a random Gaussian matrix with i.i.d. $\mathcal{N}(0,1)$ entries and write $A = \begin{bmatrix} X \\ Y \end{bmatrix}$, where $X \in \mathbb{R}^{\ell \times \ell}$ and $Y \in \mathbb{R}^{(d-\ell) \times \ell}$. Then $U^{\top}Q$ has the same distribution as $X(A^{\top}A)^{-1/2}$. It follows that $\sigma_{\min}(U^{\top}Q)$ has the same distribution as $\sigma_{\min}(X(A^{\top}A)^{-1/2})$. Note that

$$\sigma_{\min}\left(X(A^{\top}A)^{-1/2}\right) \geq \sigma_{\min}(X)\sigma_{\min}\left((A^{\top}A)^{-1/2}\right) = \frac{\sigma_{\min}(X)}{\sigma_{\max}(A)}.$$

In view of [Ver10, Corollary 5.35], $\sigma_{\max}(A) \lesssim \sqrt{d} + \log(1/\epsilon)$ with probability at least $1 - \epsilon$. Moreover, in view of [Sza91, Theorem 1.2], $\sigma_{\min}(X) \geq \epsilon/\sqrt{\ell}$ with probability at least $1 - O(\epsilon)$. The desired conclusion readily follows. $\qquad\square$