# OpenReview forum: "Global Convergence of Federated Learning for Mixed Regression"
_NeurIPS.cc/2022/Conference — NeurIPS 2022 Accept_

### Official Review · Reviewer_777o · 2022-06-26

**Rating:** 5
**Confidence:** 3
**Soundness:** 3 good
**Presentation:** 2 fair
**Contribution:** 3 good

**Summary:**

The paper studies the mixture linear regression problem under the federated learning setting. One major contribution is that the paper studies the situation where when clients’ local data volume is highly unbalanced. The authors introduce a two-phase estimation procedure. The Phase 1 aims to estimate the parameters of anchor clients (clients with large data volume), and the Phase 2 then gets the estimations of all clients by using the anchor clients’ estimation results obtained from the Phase 1. Theoretical guarantees are developed to support their methodologies.

**Questions:**

May the authors propose some experimental design to justify their algorithms?

**Limitations:**

The authors have adequately addressed the limitation concerns.

**Strengths And Weaknesses:**

Originality: The method and theory are both novel to me. Typically, in Phase 1, the author utilizes clients with low data volume to help reduce the sample complexity of estimating the parameters of anchor clients, which is clever. Besides, the theoretical results are highly nontrivial.

Quality: The mythology and theoretical analysis are in good quality. Especially, the development of the theoretical results is highly nontrivial. However, I am not familiar with relevant literature on mixture linear regression, thus I cannot judge the significance of the contribution made by the paper on theoretical perspectives.

One major concern, however, is that there is no experiment in the paper. Given that the paper is proposing new algorithms, it is not ideal to miss the empirical evidence. I understand that the main contribution of the paper is on theory; but I personally think that empirical evidence is still necessary.

Clarity: The paper is in general clear. I can follow the main ideas of the paper. However, I think there should be more discussion on the motivation of the design of Federated Moment Descent (FedMD) algorithm. The current description is not clear enough for me to follow easily.

Significance: Both methodologies and theories are significant. However, the lack of experiments hurts the significance of the paper in all.

---

> ### Author Response · Authors · 2022-08-02
> **Motivation of FD + technical contributions + experiments**
>
> **(1) Motivation of Federated Moment Descent**
>
> We will include a detailed discussion of the motivation
> behind our Federated Moment Descent algorithm. In fact, as shown in Appendix C, in the special case with two symmetric clusters and standard Gaussian features, we can obtain coarse estimation in Phase 1 via a simple power method. Unfortunately, this simple power method no longer works for the more general case with multiple clusters and sub-Gaussian features, due to the following two-fold reasons. First, with sub-Gaussian features, the leading eigenspace of $Y$ may not align with the
>  space spanned by the true model parameters $(\theta^*_1, \ldots, \theta^*_k)$. Second,
>  even if they were aligned, there would still be
>  significant ambiguity in determining the
>  model parameters from their spanned subspace. To overcome these challenges, we leverage the data heterogeneity in FL systems and design the Federated Moment Descent algorithm.
>
> **(2) Significance of the theoretical contribution in mixture linear regression**
>
>  Departing from standard mixed regression, in which each client keeps one data point only, in our problem the sizes of local datasets can vary significantly across clients. A similar mixed regression setup with data heterogeneity has been considered in [KSS+20] in a different context of meta-learning; the focus therein is on exploring structural similarities among a large number of tasks in centralized learning. On the technical side, their analysis only works when the covariance matrices of all the clusters are identical and each client has $\Omega(\log k)$ data points. Please refer to Remark 1 for more detailed technical comparisons.
>
>
> **(3) On your suggestions on adding experiments**
> For this review response, we present some preliminary in Appendix H of our revised supplementary material, which will be refined and enriched in a final version.
>
> We are running experiments to provide numerical validation of our two-phase algorithms and are in the process of polishing up our codes. Until now, we managed to obtain a collection of preliminary experimental results on synthetic data, covering multiple representative setups including unbalanced local dataset sizes. Overall, the preliminary results look promising. We will continue enriching our experiment setups. Before that, if time permits, we are more than happy to provide results under more general setups and real data upon request.
>
> In our experiments, we started with the two-symmetric cluster special case that we presented in Appendix C, followed by multiple clusters with a focus on $k=3$ for now. We choose standard Gaussian features with d=100 (feature dimension), and
> the observation noise $\zeta \sim 0.2\mathcal{N}(0, 1)$, i.e., $\sigma=0.2$.  We run Phase 1 for 50 iterations, followed by Phase 2 FedX+clustering for 500 iterations. For Phase 2, we test FedProx and FedAvg with per-iteration local update steps $s=1, 3, 5$, respectively.
> The three configurations with increasing heterogeneity in the client population are considered: (1) balanced local data, (2) unbalanced local data, and (3) unbalanced local data and unbalanced clusters (i.e., each cluster has a different number of clients in expectation).
>
> The overall error (metric formally defined in Eq.(6)) tendency against training iterations is: During the first 50 iterations (Phase 1), the errors quickly (exponentially with a large rate) drop to a plateau and remain around the same error level till the end of Phase 1. Starting from iteration 51 (upon entering Phase 2), the errors decay exponentially fast (with a smaller rate than Phase 1). These observations are consistent with our theoretical predictions.

---

> > ### Comment · Reviewer_777o · 2022-08-03
> > **Thanks for the Response**
> >
> > I would like to thank the authors for making the response. I have read the rebuttal and the revised appendix. The added simulation study in Appendix seems promising. I have two follow-up questions: 1. Would you please suggest any comparators? 2. Would you please suggest any real-world dataset for practical interest? Thanks.

---

> > > ### Author Response · Authors · 2022-08-05
> > > **Comparators and datasets**
> > >
> > > We thank the reviewers for carefully reading our rebuttal and revised appendix.
> > > See below for our responses to the two follow-up questions.
> > >
> > > ### Suggestion of comparators
> > > There are a number of algorithms that can be compared with our two-phase methods.
> > >
> > > - **Vanilla FedAvg and FedProx (training a common model)** These two algorithms are widely-adopted in practice. Under these algorithms, only a common model is trained but is used for local inference tasks (i.e., prediction) for all clients, ignoring the heterogeneity in the underlying true models across clients.
> > > - **MOCHA [SCST17]** MOCHA is the first federated multi-task learning algorithm. To encode the cluster structure of local true models, we would like to adopt the regularization term given in Eq.(11) in Appendix B.1 of [SCST17]. Since the clustering structure needs to be learned, the objective is non-convex. Convex relaxation is needed.
> > > - **Iterative Federated Clustering Algorithm (IFCA) [GCYR20]**  This algorithm uses random initialization followed by iterative clustering. We have tested our algorithm from random initialization in place of Federated Moment Descent and found that the performance is much worse.
> > > - **One-shot clustering [GHYR19]** In this algorithm, each client first sends its locally estimated model to PS, then PS clusters the clients based on the received local model estimates, and finally PS runs the outlier-resilient decentralized optimization on each estimated cluster. In the presence of highly unbalanced local dataset sizes with a large number of clients having small a local data volume (much smaller than the model dimension), the clustering errors can be high, impairing the accuracy of the trained models.
> > >
> > > Below are other simple comparators:
> > > - **Local model** Each client estimates its own local model without joining federated learning.
> > > - **Oracle clustering** The underlying true cluster is assumed to be known a priori. FedAvg or FedProx algorithm is executed on each cluster.
> > > - **Centralized mixed regression** PS has access to all the local data and runs a state-of-the-art centralized mixed regression method.
> > >
> > > ### Suggestion of real datasets
> > > We have found a number of real datasets across different application domains for our experimental study.
> > >
> > > - There are a couple of realistic federated datasets available under [LEAF](https://leaf.cmu.edu/) [CDW+19]. For example, Federated Extended MNIST (FEMNIST) consists of handwritten digits or letters from many different writers in which the writing styles of different people may be clustered. To apply our two-phase method to this dataset, we can adopt the random feature model [RR07] and run the mixed least squares regression on the random features.
> > > - There are also several real datasets for mixed regression that we can adapt to the federated setting, such as pharmacy [Sch09, Section 4.7], business, [KC07, Section 7], and music [VT02, Section 3].
> > >
> > > ### Additional references
> > > - [CDW+19] Caldas, S., Duddu, S. M. K., Wu, P., Li, T., Konečný, J., McMahan, H. B., Smith V., & Talwalkar, A. (2019). LEAF: A Benchmark for Federated Settings. Workshop on Federated Learning for Data Privacy and Confidentiality.
> > > - [KC07] Khalili, A., & Chen, J. (2007). Variable selection in finite mixture of regression models. Journal of the american Statistical association, 102(479), 1025-1038.
> > > - [RR07] Rahimi, A., & Recht, B. (2007). Random features for large-scale kernel machines. Advances in neural information processing systems, 20.
> > > - [Sch09] Schlattmann, P. (2009). Medical applications of finite mixture models. Berlin: Springer.
> > > - [VT02] Viele, K., & Tong, B. (2002). Modeling with mixtures of linear regressions. Statistics and Computing, 12(4), 315-330.

---

> > > > ### Comment · Reviewer_777o · 2022-08-08
> > > > **Reply to Author Response**
> > > >
> > > > Thanks the authors for suggestiing potential comparators and real datasets. Given that the experimental design seems convincing, I would like to raise my score to boarderline acceptance. I cannot make strong recommendation for acceptance since we cannot see the complete experiment. I hope that the authors could make more careful progress in completing the experiments if the paper is accepted.

---

> > > > > ### Author Response · Authors · 2022-08-09
> > > > > **Progress on algorithm comparison experiments**
> > > > >
> > > > > Thank you for your positive feedback! We have managed to compare the performance of our two-phase algorithm with existing FL algorithms. We presented the results in our supplementary materials as a new subsection in Appendix H. We will continue polishing up our codes.
> > > > >
> > > > > In the added subsection, we compare the performance of our two-phase algorithm with (1) vanilla FedAvg, (2) IFCA, and (3) oracle iterative clustering. We use oracle iterative clustering as a benchmark. It is an ideal implementation of our algorithm and IFCA with the estimates of the cluster models to be initialized with their ground truth. Clearly, the oracle iterative clustering is infeasible in practice.
> > > > >
> > > > > For each of the methods, we choose FedAvg as the local model update rule with s=5. We randomly initialize our two-phase algorithm, vanilla FedAvg, and IFCA. We test the performances of those algorithms under the same synthetic data and the experimental setups with the three client population configurations as adopted in Section H.2. The detailed comparisons are given in figures (c.1), (c.2), and (c.3). The plots show that our method outperforms vanilla FedAvg and IFCA, and quickly converges to the same estimation error attainable by the oracle algorithm. Note that FedAvg does not converge to small errors due to lack of model personalization in the presence of model heterogeneity. The performance of IFCA is highly dependent on the quality of initialization. With a random initialization, IFCA either suffers from a large final estimation error (see (c.1)) or gets stuck on an error floor for a long time (see (c.2) and (c.3)). In contrast, Phase 1 of our method converges rapidly (the small error fluctuation is due to the fact that we fix the number of rounds in Phase 1 to be 50).

---

### Official Review · Reviewer_DfZZ · 2022-07-08

**Rating:** 7
**Confidence:** 4
**Soundness:** 4 excellent
**Presentation:** 3 good
**Contribution:** 3 good

**Summary:**

This paper generally studies the problem of clustered federated learning. Specifically, the paper proposes a method for clustered federated learning under a mixed regression setting. At its core, the paper explains the method (dividing it into an initial estimation phase and a fine-tuning phase) and gives convergence guarantees about the statistical error of incurred by the method. The theoretical proofs are the bulk of the paper's contribution, and involve eigengap-free bounds on subspace estimation and VC dimension analyses of certain classes of polynomials.

**Questions:**

1. The application of the Milnor-Thom theorem in the proof of Lemma 10 is not quite correct. The bound quoted on the number of sign patterns in L684 of the appendix is only valid if $m(k-1) \geq dk$. However, if this does not hold then the resulting bound on $m$ follows anyways, so the lemma is still true. However, this is worth cleaning up.

2. The assumption that $\phi(x_{ij})$ is sub-Gaussian means that the assumption that $\phi(\cdot)$ can be a non-linear mapping is not particularly useful, and simply adds unnecessary notation. I think the paper would be clearer if $\phi$ was omitted altogether, and the authors assume a mixed linear regression setting with sub-Gaussian data.

3. Intuitively, what does the bound on $\nu$ in the assumptions of Theorem 1 impose?

**Limitations:**

There are no negative societal impacts of the work. My only concern is whether or not the method proposed can be made private (in some formal notion) and this is discussed above.

**Strengths And Weaknesses:**

## Core Strengths

The core contribution of the paper (as it is claimed in the paper) is novel and exciting: Rigorous guarantees on clustered federated learning are few and far between, and I believe that trying to attain this is a significant strength of the paper. Moreover, I think the authors do a good job of motivating this type of approach and giving a rigorous taxonomy of the related literature.

At a high level, I believe that the paper's theoretical results are significant and enough to warrant acceptance. It is clear that there is a lot of interesting facets to the analysis, and the authors mainly do a good job of blending together a variety of disparate analytical viewpoints. In particular, the strategy outlined to prove Theorems 1 & 2 seems promising and mainly correct. I have not verified every detail though, but the portions I have looked at all seem correct. In particular, I especially appreciated the application of the Milnor-Thom theorem (though there is a slight, fixable issue with that, see the suggestions below).

## Weaknesses

The biggest weakness of the paper (and the method proposed by the authors more generally) is its questionable pertinence to federated learning. In particular, as the authors state in their "Broad Impact" section, one of the primary tenets of federated learning is the ability to perform machine learning without compromising data privacy. In the original FedAvg paper, for example, the models trained by clients are never shared directly with one another or the server, but are instead all averaged together (which paves the way for things like differential privacy and secure aggregation, which have formal privacy guarantees).

However, Algorithm 1 repeatedly sends models that are held by a client (specifically an anchor client) to many other clients. While this is not the same as broadcasting data directly, it seems questionable as to whether this method is preserving data privacy. This is particularly concerning due to the linear structure of the problem, where clients could conceivably use an anchor client's model to learn about the client's data. Note that I do not believe that all FL algorithms need to incorporate privacy from the start. However, I think that a lack of any reasonable pathway to privacy integration is a drawback to a method that should be discussed and considered. As a naive reference point, an algorithm that had clients send all their data directly to one another would be effectively useless as an FL method.

This is all to say that I believe that the paper should 1) be up front about what kind of privacy model it is actually envisioning and 2) discuss possible pathways to improving the privacy of the algorithm (if there are any). If the authors believe that broadcasting cluster model estimates (as in Algorithm 1, Line 9) directly to other clients is acceptable, then I believe that this warrants discussion.

A more minor weakness of the work is that some of the notation becomes cumbersome, and relies on quantities/assumptions that could use extra explanation. For example, Theorem 2 makes an assumption on the "clustering error" term $\nu$ (namely, k$\nu\log(e/\nu) \leq \rho/\kappa$). It is not immediately clear how strong or weak this assumption is. If I understand correctly, then this actually imposes a restriction on the dimension $d$, which seems like an important thing to note.

## Summary

I think this is a theoretically strong paper that studies an important topic in federated learning. However, I would prefer that the authors are more up front about the actual privacy model underlying their method, and whether or not that has a pathway to integration with privacy mechanisms like differential privacy or secure aggregation.

### Review Score

I have opted to only give the work a weak accept (6) at the time of writing this review. While I believe the theory is strong, this score is tied to the privacy concerns above. I would be happy to revisit this score, contingent upon the work including a useful, grounded discussion of the privacy risks of the method and potential pathways forward.


# Concerning the Author Feedback and Discussion

This section has been added after reading the other reviews and author feedback. The authors have addressed my feedback well, and have even gone so far as to add entirely new experiments (as per the request of another reviewer). I do not believe that these were necessary for the purposes of acceptance, but cement my belief that this paper should be accepted. I have increased my score from a 6 to a 7 correspondingly.

---

> ### Author Response · Authors · 2022-08-02
> **Response to privacy issues and other questions**
>
> We would like to thank you for your positive feedback and constructive suggestions.
>
> (1) **On data privacy:** Thank you very much for pointing this out. We will add detailed discussion on privacy protection of our methods in the revised manuscript. In short, our algorithm is fundamentally more private than the reference point algorithm mentioned in the review (i.e., the algorithm that had clients send all their data directly to one another). Compared to the standard FedAvg algorithm wherein only aggregated local updates/gradients are broadcasted by the parameter server, the major step of our two-phase algorithm that may leak additional privacy is Step 8 wherein
> the local model estimates of the anchor clients are broadcasted to many other non-anchor clients.
> However, this privacy leakage is minor and can be further mitigated by
> a simple privacy-preserving mechanism.
>
>
> - First: In our algorithm, each chosen non-anchor client only receives a collection
> of local model estimates (without ID for anchor clients) from the parameter server, it does not know which broadcasted model
> corresponds to which anchor client and hence cannot directly identify each individual anchor client's local true model.
>
> - Second: We only choose a very few number of anchor clients (roughly on the order of the number of clusters)
> and in practice these anchor clients are often specially recruited by the PS; hence they can be made
> less concerned about privacy leakage through some incentivizing schemes.
>
> - Last but not least, we can better preserve the privacy of anchor clients by broadcasting perturbed versions of their local
> models to each client. Specifically, fix any anchor client $i$,
> each non-anchor client $i'$ receives $\theta_{i',i,t}$
> and $\tilde \theta_{i',i,t}$ that are equal to $\theta_{i,t}$ subject to
> two independent noise perturbations. Then for the subspace estimation in Step 9, we can replace two $\theta_{i,t}$'s,
> one by $\theta_{i',i,t}$ and the other by $\tilde \theta_{i',i,t}$, in the definition of $Y_{i,t}.$ Crucially, $Y_{i,t}$ involves an average over $m$ non-anchor clients $i'$; hence these independent noise perturbations for different $i'$ will be averaged out. Since $m$ is large, this implies that the injected random noises can be made large without deteriorating too much the accuracy of the subspace estimation, in a similar spirit as privatizing the model averaging step in FedAvg.
> **This gives a  promising pathway to maintain anchor clients' privacy**.
>
> We will include the current discussion in the revision as suggested.
> However, we feel that incorporating this privacy-preserving mechanism into our method and rigorously analyzing its
> privacy guarantee are beyond the scope of this work, so with your permission, we would rather leave this as future work.
>
>
>
> (2) **(Using $\phi(x)$ under sub-Gaussian assumption)**
>
> We were aware of this and agree that mathematically it is cleaner to omit the non-linear transformation $\phi$ and focus on a mixed linear regression setting with sub-Gaussian data.
> However, we are worried that omitting $\phi$ will give readers the wrong impression that our setup can only handle the restricted linear regression. In contrast, with a bit increased yet manageable notation complexity,   explicitly showing $\phi$, our setup covers polynomial regression and can be further used to approximate neural networks in certain regimes through neural tangent kernels, etc.
>
> (3) **(Assumptions on $\nu$)**
>
> We will add further clarifications on the assumptions imposed on the clustering error $\nu$.
> As we explained right after Theorem 2, the clustering error $\nu$ consists of two parts shown in eq. (10): the first term  decays exponentially in the local dataset size and the signal-to-noise ratio and hence is very small in most typical scenarios;
> the second term is on the order of $dk \log k/M$ when the quantity skewness (imbalance of data partition) is of a constant order. Finally, $\rho$ is a constant in many applications, and we can choose a small enough step size to ensure $\kappa\approx 1$. Thus the key assumption $\nu\log(e/\nu) \le \rho/\kappa$ roughly translates to $\nu$ being a subconstant,
> which further means that
> $M$ (the number of clients) needs to be larger than $d$ (the model dimension) by polylog factors. This is often satisfied in the typical FL applications which involve a very large collection of clients.
>
>
> (4) **(Application of the Milnor-Thom theorem)**
>
> Thank you for catching this. Indeed, to apply the Milnor-Thom theorem, we need to satisfy the condition $m(k-1) \ge dk$. If this condition is not satisfied, that is $m(k-1)<dk$, then our desired bound $m \lesssim dk \log k$ holds automatically. We will clarify this in the revision as suggested.

---

> > ### Comment · Reviewer_DfZZ · 2022-08-04
> > **On discussions of privacy in federated learning papers**
> >
> > Thank you for the detailed response. Just to clarify, I agree with the authors (and tried to make this clear in my review) on the following points:
> >
> > 1. The algorithm proposed in this work is more private than the naive reference point I sketched (wherein all clients share all data).
> > 2. It is not necessary for every FL paper to have explicit differential privacy integration. Rather, I think it is important to at least acknowledge possible privacy violations, and discuss how future work may address them.
> >
> > The discussion you give above is essentially what I was recommending be added to the paper: Discussion of how privacy is compromised (or not compromised), what models of privacy this could be used under (eg. a model where the anchor clients are incentivized to make their models public), and possible integrations with formal privacy mechanisms. However, I would be careful about making unsubstantiated claims about whether an approach will or will not work.
> >
> > Regarding the use of $\phi$: This is a minor point, but perhaps this could be the subject of a comment, rather than carrying through the notation throughout the work? Eg. Some kind of paragraph or footnote saying that this is still appropriate to polynomial regression and NTK settings, though you omit the mapping $\phi$ for clarity purposes.
> >
> > Regardless, I think you have basically addressed my points above. Thanks.

---

### Official Review · Reviewer_SAAF · 2022-07-11

**Rating:** 7
**Confidence:** 4
**Soundness:** 3 good
**Presentation:** 3 good
**Contribution:** 3 good

**Summary:**

The following points summarize the paper:

1. The authors propose a two-phase federated learning algorithm to learn cluster label and mixed regression parameter vectors.
2. In Phase 1, few anchor clients (with $\tilde{\Omega(k)}$ data) obtain a course model estimate.
3. In Phase 2, each client alternately estimates its cluster label and refines the model estimates based on FedAvg or FedProx.
4. Their method works with unbalanced data across clients.
5. They also provide global convergence of their algorithm with any initialization.


**Questions:**

- Is $R$ (and $\Delta, \sigma$) treated as a constant in the proofs? If not, does it change the theoretical results (concentration rate etc)?

**Limitations:**

The limitations are adequately addressed by the authors.

**Strengths And Weaknesses:**

Strengths:
Authors have done a commendable job of providing theoretical gurantees for their claims. The paper is theoretical in nature and like any theoretical paper, seems a bit tedious to parse at times. However, on a high level all the proofs seem to follow naturally and support the claim. Overall, it is a well written paper with strong theoretical claims.

Weaknesses:
- The assumptions such $\| \theta_j^* \|  \leq R$ seem quite strong as they essentially remove $O(\sqrt{d})$ dependency (assuming $R$ is treated as constant) from the problem. This combined with conditions like $\frac{R}{\Delta} = \mathcal{O}(1)$ seem like quite stringent conditions on the entries of $\theta_j^*$.
- Although, it is a theoretical paper, it would have been nice to add some experiments (on synthetic or real data) to validate the theorems.

---

> ### Author Response · Authors · 2022-08-02
> **On the scaling of $R, \Delta, \sigma $ and experiments**
>
> We first would like to thank you for your positive feedback, constructive suggestions, and question that you raised.
>
> (1) On your question *"Is  $R, \Delta, \sigma $ treated as a constant in the proofs?"*:
>
> **Our response:**
> We do not treat $R, \Delta, \sigma $ as constants. In our analysis, we allow $R$ (similarly for $\Delta$ and $\sigma$) to scale with $d$, e.g., $R=O(\sqrt{d})$. In fact, if they are constants, we do not even need to impose the assumption that $R/\Delta=O(1)$ since it trivially holds.
>
> The assumption $R/\Delta=O(1)$ is imposed for ease of presentation only -- as we briefly mentioned in Section 3 right before the notation paragraph. Our results can be extended to show more explicit dependencies on the ratio $R/\Delta$ with careful bookkeeping calculations. Also, we note that the assumption $R/\Delta=O(1)$ is reasonable, as it basically requires the radius of $\theta_j^*$s is on the same scale as the pairwise separation. It only rules out the very extreme setting where $\theta_j$s themselves are extremely large while their pairwise separations are tiny. We will add clarifications on relevant scalings of $R,\Delta, \sigma$ in our revision.
>
> (2) On your suggestions on adding experiments:
>
> **Our response:**
> The plots can be found in Appendix H of our revised supplementary material.
>
> We will add comprehensive experiments covering a variety of FL heterogeneity setups in our revised manuscript.
> We are running experiments to provide numerical validation of our two-phase algorithms and are in the process of fine-tuning and polishing up our codes. Until now, we managed to obtain a collection of preliminary experimental results on synthetic data, covering multiple representative setups including unbalanced local dataset sizes. Overall, the preliminary results look promising, and the plots can be found in our revised manuscript Appendix H. (We put the plots in the appendix as we did not find an alternative way to post plots in the response.) We will continue enriching our experiment setups and, if time permits, provide results under more general setups and real data upon request.
>
> We run Phase 1 for 50 iterations, followed by Phase 2 FedX+clustering for 500 iterations. For Phase 2, we test FedProx and FedAvg with per-iteration local update steps $s=1, 3, 5$, respectively. The three configurations with increasing heterogeneity in the client population are considered: (1) balanced local data, (2) unbalanced local data, and (3) unbalanced local data and unbalanced clusters (i.e., each cluster has a different number of clients in expectation).
>
> The overall error (metric formally defined in Eq.(6)) tendency against training iterations is: During the first 50 iterations (Phase 1), the errors quickly (exponentially with a large rate) drop to a plateau and remain around the same error level till the end of Phase 1. Starting from iteration 51 (upon entering Phase 2), the errors decay exponentially fast (with a smaller rate than Phase 1). These observations are consistent with our theoretical predictions.

---

> > ### Comment · Reviewer_SAAF · 2022-08-05
> > **Response to rebuttal**
> >
> > Thank for your clarification on the scaling of $R, \Delta$ and $\sigma$. Your initial experimental results are promising and it will be great if you could add more comprehensive experiments.

---

### Meta-Review · Area_Chair_NUCA · 2022-08-30

**Recommendation:** Accept
**Confidence:** Certain

**Metareview:**

This is mainly a theoretical piece of work studying the problem of clustered federated learning under a mixed regression setting. The authors establish convergence guarantees related to the statistical error incurred by the method; the results include eigengap-free bounds on subspace estimation and VC dimension analyses of certain classes of polynomials.

The reviewers highlighted many positive attributes of the work, including:

- The core contribution of the paper (as it is claimed in the paper) is novel and exciting: Rigorous guarantees on clustered federated learning are few and far between, and I believe that trying to attain this is a significant strength of the paper.
- I think the authors do a good job of motivating this type of approach and giving a rigorous taxonomy of the related literature.
- At a high level, I believe that the paper's theoretical results are significant and enough to warrant acceptance. It is clear that there is a lot of interesting facets to the analysis, and the authors mainly do a good job of blending together a variety of disparate analytical viewpoints. In particular, the strategy outlined to prove Theorems 1 & 2 seems promising and mainly correct. I have not verified every detail though, but the portions I have looked at all seem correct. In particular, I especially appreciated the application of the Milnor-Thom theorem (though there is a slight, fixable issue with that, see the suggestions below).
- I think this is a theoretically strong paper that studies an important topic in federated learning.
- The method and theory are both novel to me. Typically, in Phase 1, the authors utilize clients with low data volume to help reduce the sample complexity of estimating the parameters of anchor clients, which is clever. Besides, the theoretical results are highly nontrivial.
- The methodology and theoretical analysis are of good quality. Especially, the development of the theoretical results is highly nontrivial.
- The paper is in general clear. I can follow the main ideas of the paper.
- Both methodologies and theories are significant.

Besides these words of praise, some criticism was mentioned. The remaining criticism (after rebuttal and discussion) was not very major, however. **I therefore, and with pleasure, propose to accept this paper.** I would wish to stress that it is important to properly address all issues in the camera-ready version of the paper.

Best regards,

AC

**Award:**

No

---

### Decision · Program_Chairs · 2022-09-14

Accept